# Pyruvate Dehydrogenase A1 Phosphorylated by Insulin Associates with Pyruvate Kinase M2 and Induces LINC00273 through Histone Acetylation

**DOI:** 10.3390/biomedicines10061256

**Published:** 2022-05-27

**Authors:** Abu Jubayer Hossain, Rokibul Islam, Jae-Gyu Kim, Oyungerel Dogsom, Kim Cuong Cap, Jae-Bong Park

**Affiliations:** 1Department of Biochemistry, Hallym University College of Medicine, Chuncheon 24252, Kangwon-do, Korea; yubayerbge89@gmail.com (A.J.H.); rakibbgeiu@yahoo.com (R.I.); mip11@hallym.ac.kr (J.-G.K.); oongee.oyungerel@gmail.com (O.D.); kimcuong.cap@gmail.com (K.C.C.); 2Institute of Cell Differentiation and Aging, Hallym University College of Medicine, Chuncheon 24252, Kangwon-do, Korea; 3Department of Biotechnology and Genetic Engineering, Faculty of Biological Science, Islamic University, Kushtia 7003, Bangladesh; 4Department of Biology, School of Bio-Medicine, Mongolian National University of Medical Sciences, Ulaanbaatar 14210, Mongolia; 5Institute of Research and Development, Duy Tan University, Danang 550000, Vietnam

**Keywords:** p-PDHA1, PKM2, insulin, histone acetylation, LINC00273

## Abstract

Insulin potently promotes cell proliferation and anabolic metabolism along with a reduction in blood glucose levels. Pyruvate dehydrogenase (PDH) plays a pivotal role in glucose metabolism. Insulin increase PDH activity by attenuating phosphorylated Ser293 PDH E1α (p-PDHA1) in normal liver tissue. In contrast to normal hepatocytes, insulin enhanced p-PDHA1 level and induced proliferation of hepatocellular carcinoma HepG2 cells. Here, we attempted to find a novel function of p-PDHA1 in tumorigenesis upon insulin stimulation. We found that p-Ser293 E1α, but not the E2 or E3 subunit of pyruvate dehydrogenase complex (PDC), co-immunoprecipitated with pyruvate kinase M2 (PKM2) upon insulin. Of note, the p-PDHA1 along with PKM2 translocated to the nucleus. The p-PDHA1/PKM2 complex was associated with the promoter of long intergenic non-protein coding (LINC) 00273 gene (LINC00273) and recruited p300 histone acetyl transferase (HAT) and ATP citrate lyase (ACL), leading to histone acetylation. Consequently, the level of transcription factor ZEB1, an epithelial–mesenchymal transition (EMT) marker, was promoted through increased levels of LINC00273, resulting in cell migration upon insulin. p-PDHA1, along with PKM2, may be crucial for transcriptional regulation of specific genes through epigenetic regulation upon insulin in hepatocarcinoma cells.

## 1. Introduction

Insulin has a variety of functions that include stimulation of glucose uptake and glycogen synthesis and biosynthesis of proteins and lipids in various cells. In addition, insulin increases the expression of several target genes to stimulate cell proliferation and anabolic actions [1]. Insulin signals via binding to the insulin receptor with signal propagation through the insulin receptor substrate (IRS), phosphatidyl inositol-3-kinase (PI3K), Akt/PKB, glycogen synthase kinase-3β (GSK-3β), and mTOR [2,3]. It is well established that activated Akt induces phosphorylation at Ser9 of GSK3β (p-Ser9 GSK-3β) [4,5].

Oxidation of pyruvate is the necessary first step in mitochondria to produce acetyl-CoA and CO_2_. Acetyl-CoA then participates in the TCA cycle to produce NADH + H^+^ in the mitochondrial matrix and generate ATP. For pyruvate oxidation, pyruvate dehydrogenase complex (PDC) is a key regulator. PDC is composed of PDH E1, dihydrolipoyl transacetylase E2, and dihydrolipoyl dehydrogenase E3 subunits and PDC is mainly regulated by the phosphorylation of its PDH E1α subunit (PDHA1) at Ser293 residue in mouse and Ser264 residue in human though PDH kinases and phosphatases [6]. Insulin is known to activate PDH activity and attenuate p-PDHA1 levels in normal liver tissue and hepatocytes. Furthermore, post-translational modifications such as succinylation and metabolites, including acetyl-CoA/CoA, NADH/NAD^+^, and ATP, regulate PDC activity [7]. 

Pyruvate kinase (PK) is the other critical enzyme in the glycolysis process; it catalyzes the final step in converting phosphoenol pyruvate (PEP) and ADP to pyruvate and ATP. PKM1 and PKM2 are expressed by alternatively splicing the PKM1/2 gene. It is noteworthy that PKM2 is highly expressed in embryo and cancer tissues, whereas PKM1 is only expressed in normal adult tissues [8]. Notably, PKM2 plays a role as a transcription factor [9] to regulate the expression of specific genes such as glucose transporter 1 (GLUT1) and lactate dehydrogenase A (LDHA) [10], and cyclin D1 [11]. PKM2 also functions as a protein kinase; it phosphorylates histone 3 (H3) Th11 upon EGF receptor activation, leading to the expression of cyclin D1 and c-Myc and tumor cell proliferation [12]. PKM2 also phosphorylates Stat3 at Tyr705 as one of its targets [13].

Recently, we reported that insulin induces an increase in p-Ser293 PDHA1 along with p-Ser473 Akt levels in HepG2 hepatocellular carcinoma cells. However, insulin reduced p-Ser293 PDHA1 in insulin-treated rat liver tissue and rat primary hepatocyte [14]. However, the role of p-PDHA1 in hepatocellular carcinoma HepG2 cells due to insulin is poorly understood. Herein, it was necessary to study the effect of insulin on HCC cells such as HepG2 cells as a model system with regard to PDHA1 Ser293 phosphorylation.

In this study, we found that p-PDHA1 binds to PKM2, and both are localized to the nucleus in HepG2 cells in response to insulin. We observed p-PDHA1/PKM2 complex regulates the expression of long intergenic non-protein coding RNA 00273 (LINC00273), though recruiting p300 histone acetyltransferase (HAT) and ATP citrate lyase (ACL), concomitant with the regulation of histone acetylation in the promoter of LINC00273.

## 2. Materials and Methods

### 2.1. Materials

Nonidet P-40 (NP-40), bovine serum albumin (BSA), poly-L-lysine solution (P8920), dichloroacetic acid (DCA), SB-415286, and isopropyl β-D-thiogalactoside (IPTG) were purchased from Sigma-Aldrich Co. (St. Louis, MO, USA). Y27632 was obtained from Calbiochem (La Jolla, CA, USA). Dulbecco’s modified Eagle’s medium-F12 (DMEM-F12), fetal bovine serum (FBS), penicillin, and streptomycin were purchased from Cambrex (Verviers, Belgium). LB Broth High Salt (MB-L4488) and skim milk powder (MB-S1667) were from MBcell (SeoCho-Gu, Seoul, Korea). Methanol-free formaldehyde was obtained from Pierce (Rockford, IL, USA). Alexa flour 488 goat anti-mouse IgG, 4′6-diamidino-2-phenylindole (DAPI), and Lipofectamine 3000 were from Invitrogen (Carlsbad, CA, USA). ProLong Gold Antifade mounting solution, Alexa flour-568, and Alexa flour-594 reagents were obtained from Molecular Probes (Eugene, OR, USA). Polyvinylidene difluoride (PVDF) membrane was purchased from Millipore (Billerica, MA, USA). Attractene (301005) and Hyperfect (301802) transfection reagents were from Qiagen (Valencia, CA, USA). jetPRIME DNA/si-RNA transfection reagent was obtained from Polyplus-transfection (Seoul, Korea). Human recombinant insulin (Cat: INSL16-N-5) was purchased from Alpha Diagnostic International (ADI) (San Antonio, TX, USA). Glutathione (GSH)-sepharose 4B/agarose and protein A/G-agarose beads were from Amersham Biosciences (Piscataway, NJ, USA).

Following antibody were used or this study: normal IgG (sc:2025, Santacruz and 2729S Cell Signaling Technology, Danvers, MA, USA), p-PDHA1 (ab92696, Abcam, Cambridge, UK), PKM2 (4053, Cell Signaling Technology, JM001 and 32054, Signalway Antibody), PKM1 (21577, Signalway Antibody, Greenbelt, MD, USA), PDHA1 (sc:377092, Santacruz), p-PKM2 (3837, Cell Signaling Technology), actin (sc:58673, Santacruz), PDH E2 (ab126224, Abcam), PDH E3 (ab133551, Abcam), Ac-H3 k27 (ABP50203, Abbkine, CA, USA, Ac-H3 K18 (ABP50201, Abbkine), Ac-H3 k9/14 (9677, Cell Signaling Technology), p300 (bs-6954R, Bioss antibodies), ATP citrate lyase (ACL) (sc:517267, Santacruz), HIF-1α (3176, Cell Signaling Technology), Oct4 (GTX-627419, Gene Tex), Stat3 (9139, Cell Signaling Technology), Lamin B (sc:965962, Santacruz), Tubulin (sc:32293 and 2144, Cell Signaling Technology), HA-Tag (3724, Cell Signaling Technology and A02040, Abbkine), ZEB1 (ABP60963, Abbkine), E-cadherin (ABP0083, Abbkine), N-cadherin (sc:393933, Santacruz), vimentin (sc:6260, Santacruz), Snail (sc:271977, Santacruz), c-Myc (sc:789, Santacruz), and cyclinD1 (sc:8396, Santacruz). Finally, goat anti-rabbit and goat anti-mouse IgG conjugated with HRP were from Enzo Life Sciences (Farmingdale, NY, USA).

### 2.2. Cell Culture

Human hepatocellular carcinoma cell lines HepG2, Huh7, mouse hepatocellular carcinoma cell line Hepa1c1c7, and 4T1 mouse breast cancer cell line (Korean Cell Line Bank, Seoul, Korea) were maintained in Dulbecco’s modified Eagle’s medium (DMEM) containing high glucose, 4.5 g/L D-glucose, L-glutamine, 110 mg/L sodium pyruvate and sodium bicarbonate and PC12 cells were maintained in RPMI 1640 medium 1× with low glucose (2.0 g/L), sodium bicarbonate (2.0 mg/L), sodium chloride (6.0 g/L) supplemented with 10% heat-inactivated fetal bovine serum (FBS) with 1% penicillin/streptomycin antibiotics. Cell culture flasks were maintained under 5% CO_2_, 95% ambient air, and humidified conditions at 37 °C [14]. Experimental plate cells were first serum starvation for 12 h without serum and antibiotics and finally treated with insulin (100 nM) and other stimulants such as LPS (10 µg/mL) and NGF (100 ng/mL). There is no serum or antibiotics in all experiments.

### 2.3. Protein Extraction and Western Blot Analysis

Western blotting was conducted following the previous methods [15], and images were determined with the FUSION FX-Western Blot & Chemi & Chemiluminesent imaging system (Vilber Lourmat) [16].

### 2.4. Protein Identification by Peptide Mass Fingerprinting (PMF)

Purified recombinant GST-PDH WT, 293D, and 293A protein-containing beads mixed with PC12 cell lysate overnight at 4 °C. Protein binding to GST-PDH 293D (phospho-mimic) mutant except for GST-PDH WT and GST-PDH 293A (dephosphomimitic mutant) were identified by using MALDI-TOF analysis in Genomine Inc. (Pohang, Korea; Microflex LRF 20, Bruker Daltonics, Bremen, Germany) as described by Fernandez et al. [17].

### 2.5. Immunoprecipitation

Specific protein was immunoprecipitated with the specific antibody and proteinA/G-beads from cell lysates, and the precipitated proteins were analyzed with SDS-PAGE and Western blotting [16].

### 2.6. Cells Proliferation Measurement by MTT Reagents and DAPI Staining

HepG2 cells were seeded either in 12-well (1 × 10^5^ cells/well) or 6-well dishes (4 × 10^5^ cells/well) and 96-well plates (1 × 10^3^ cells/well). Cells were serum-starved for 12 h before adding 100 nM insulin. These media contain high glucose 4500 mg/L, L-glutamine, 110 mg/L sodium pyruvate, and sodium carbonate without serum and antibiotics. Then cells were fixed with 4% formaldehyde for 10 min at RT. The nuclei of cells were stained with 1 μg/mL DAPI (1:200) for 10 min at RT. Cell proliferation was quantified by determining the fluorescence of cell density using fluorescence microscopy [18]. In another assay, living cells were detected through colorization by using CCK-8 reagent from Quanti-Max-WST-8 cell viability assay kit (Biomax, Seoul, Korea Cat: QM 2500) and MTT reagent (M5655: Sigma-Aldrich) [16]. Finally, OD values were measured by spectrophotometer (spectramax plus384, San Jose, CA, USA) at 450 nm.

### 2.7. Recombinant GST Proteins Purification and Site-Directed Mutagenesis

PKM2 WT and domains (N, A1, B, A2, and C) were made by using PCR. EcoR1 and XhoI restriction enzyme sites included in the 5’ and 3’ ends of PKM2 gene GST-PDH WT, GST-293D, and GST-293A mutant were expressed in *E. coli BL21* using pGEX-4T1 bacterial expression vector [16]. Mutant PDHA1 S293D, PDHA1, and S293A constructs were prepared by using a site-directed mutagenesis kit (Intron Biotechnology, Sungnam, Korea) [15].

### 2.8. Transfection of Plasmid DNA and si-RNA

DNA and Small interfering RNA (si-RNAs) were transfected to cells using Lipofectamine 3000 (Invitrogen) and jetprime reagent (polyplus transfection, France) according to the manufacturer’s instructions. Various HA-pCMV-PKM2 domain and PDHA1 constructs were transfected by using Lipofectamine 3000 (Invitrogen) according to the manufacturer’s instructions. In all transfection experiments, cells were first seeded and incubated for 8 h without antibiotics but containing growth media; then media were aspirated, serum-free media without antibiotics was added for 12 h, and insulin (100 nM) was added to fresh serum-free media into the experimental plate, followed by 48 h of incubation for transfection. The media was then replaced by fresh full growth media with incubation for another 48 h before performing the stated experiments. In six-well cell culture dishes, 2 µg DNA was transfected. Small interfering RNAs (si-RNA) si-PDHA1 (sc-91064 and ID: 18597), si-PKM2 (sc-62820), and control si-RNA (sc-37007) were purchased from Santa Cruz and Bioneer. These si-RNAs were transfected at a final concentration of 30 nM-50 nm for 48 h. si-LINC00273 was purchased from Bioneer and was transfected at a final concentration of 20–50 nM for 48 h. For double transfection, first, si-RNA was transfected, then after 2 h, DNA was transfected into the cells according to the manufacturer’s protocols Lipofectamine 3000 (Invitrogen). Sometimes, we performed co-transfection for some experiments.

### 2.9. Cytosolic and Nuclear Fractions Isolation

Cytosolic and nuclear fractions were separated by using NE-PER nuclear and cytoplasmic extraction reagents (CER: Thermo Scientific, 78833). Cytosolic and nuclear proteins were analyzed by Western blotting [19].

### 2.10. Immunostaining of Cells

HepG2 cells were cultured and fixed in 4% paraformaldehyde for 10 min, and for membrane permeabilization, we used 1XPBST (containing 0.1% TX-100 detergents and PBS) for 10 min, then washed with 1XPBS and incubated with specified primary antibody (1:100) overnight at 4 °C. After primary antibody incubation, they were again washed with 1XPBS, then p-PDHA1 and PKM2 antibody recognized by an Alexa Fluor 488-conjugated secondary antibody (green-color emission) and Alexa-546 conjugated anti-rabbit IgG (red color emission, Molecular Probes) with 1:50 dilution for 2 h at room temperature. DAPI (1 μg/mL) was also added 10 min before washing to label the nuclei [16]. For dual immunofluorescence, either monoclonal anti-p-PDHA1 or anti-PKM2 at 1:100 dilution was added and incubated with the cells overnight at 4 °C. Then, we added Alexa-488 conjugated anti-mouse IgG (green color emission, Molecular Probes) with 1:50 dilution and Alexa-546 conjugated anti-rabbit IgG (red color emission, Molecular Probes) with 1:50 dilution, respectively. The nuclear region was stained with DAPI (4’,6-diamidino-2-phenylindole). Fluorescence images were obtained with a conventional fluorescence microscope (Axiovert 200, Zeiss, Oberkochen, Germany). We measured the cyan color intensity in the nucleus by using Adobe Photoshop version 7 and plotted the relative intensity as a bar diagram.

### 2.11. Monolayer Wound Healing Assay

After becoming confluent, the HepG2 cell monolayer was scratched by using a 1 mL sterile pipette tip across the center of the well. The monolayer wound area was measured by capturing and analyzing width by using a light microscope (Axiovert 200, Zeiss, and Adobe Photoshop 7). The relative fold wound closure was calculated by the following equation: wound closure = [1 − (wound area at T_t_/wound area at T_0_)]
where T_t_ is the time after wounding and T_0_ is the time immediately after wounding.

### 2.12. Chromatin Immunoprecipitation Sequencing (ChIP-Seq) and ChIP-PCR

We accomplished the ChIP-Seq by using the Abcam protocol (Abcam; Cambridge, UK). Originally, HepG2 cells were treated with insulin, then we added formaldehyde as a crosslinker (final concentration, 0.75%) for 1 h, then stopped glycin 125 mM for 15 min. After sonication, p-PDHA1 and PKM2 antibodies were incubated with the fragmented chromatin-protein complex and precipitated by protein A-beads. The beads were then washed, and the bound DNAs were eluted by the elution buffer (1% SDS and 100 mM NaHCO3). The DNA fragments were sequenced in eBiogene (Seoul, Korea). For ChIP-PCR for LINC00273, the primers were designed by Bioneer (Daejeon, Koera). The detailed information on the LINC00273 promoter was as follows (Human GRCh37/hg19) [19].

### 2.13. RNA Isolation and Quantitative Reverse Transcriptase (RT-qPCR) to Measure Target Genes’ mRNA Level

Total RNA from HepG2 cells was isolated by using TRIzol reagent (Ambion, CA, USA). The concentration and purity of the isolated RNA were measured spectrophotometrically at 260 nm and 280 nm and by Nano Drop (Thermo Scientific). cDNA from RNA was made by reverse transcription with M-MLV reverse transcriptase (NEB-UK, Hitchin, UK). A mix of 2 μg total RNA, 1 μL Oligo-dT, and 2 μL of LINC00273 primers (Bioneer, Oakland, CA, USA) was incubated in a total volume of 10 μL for 5 min at 75 °C and cooled to 4 °C for 5 min in the PCR first reaction. To the mix, 2.5 μL of M-MLV 5× reaction buffer (NEB-UK), 5 μL dNTPs, 1 μL RNAase inhibitor (NEB-UK), and 1 μL of M-MLV reverse transcriptase were added to reach a total volume of 25 μL. The reaction mixture was then incubated at 25 °C for 10 min, 42 °C for 60 min, 95 °C for 10 min and then cooled to 4 °C. For the cDNA reaction mix, PCR reactions were then performed using rTaq 5x Master Mix (ELPIS-Biotech, Daejeon, Korea) in reactions containing 10 pM of each primer, 4 μL cDNA, and 10× PCR buffer as supplied by the manufacturer in a total volume of 20 μL. PCR primers of GAPDH) were used for analysis as house-keeping control gene. The GAPDH PCR reaction was run at 95 °C for 5 min, 20 cycles of 95 °C for 15 s 60 °C for 20 s, 72 °C for 20 s, concluded with 72 °C for 5 min, and cooled to 4 °C. The cDNA was then used with the ExcelTaq SYBR Rox 2X fast Q-PCR master mix (TQ1210, SMOBIO Technology. Inc., Hinchu, Taiwan) for real-time qPCR (RT-qPCR) quantification using an Applied Biosystems Step one plus PCR system. All samples were analyzed in triplicate. Relative quantities of specifically amplified cDNA were determined using the comparative threshold cycle (CT) values method. GAPDH was used as an endogenous reference gene, and without template and reverse-transcription controls were used to exclude nonspecific amplification. For measuring LINC00273 mRNA levels, primer sequences were designed, and PCR reaction conditions were: 95 °C for 5 min, followed by 30 cycles of 95 °C for 30 s, 54 °C for 30 s, 72 °C for 30 s, concluded by 72 °C for 5 min and cooled to 4 °C. Products were run on 1% agarose gels and visualized by ethidium bromide staining. RT-PCR products were run on 1% agarose gels and visualized by ethidium bromide staining. Primers were designed for GAPDH: forward, 5′-AGAAGGCTGGGGCTCATTTG-3′, reverse, 5′-AGGGGCCATCCACAGTCTTC-3′; for LINC00273: forward, 5′-GCCACACAGTAGGTGACGAG-3′, reverse, 5′-ACTGCTTTCGGGAGAGAATG-3′; for GPR174: forward, 5′-TGTGCCAGGTCTCATAGGGA-3′, reverse, 5′-AGTCATGGAAGCGAAAGGGG-3′; for KDM1B: forward, 5′-GAGGGACAGGTGCTTCAGTT-3′, reverse, 5′-CACTGCACTGGAGATTTGAG-3′.

### 2.14. Analysis of Survival Probability of Human Liver and Lung Cancer Patients

The human liver and lung cancer tissue samples were obtained from Pusan National University Hospital (Pusan, Korea), which is supported by the National Cancer Center of Korea. We determined relative p-PDHA1 and PKM2 expression by Western blotting. Information on the death and survival of the patients was obtained from the Korean Statistical Information Service KOSTAT database (KOSIS) of the Korean government and approved by the institutional review board (IRB: HIRB-2019-048) of Hallym University (Chuncheon, Korea). When patients were non-survival or survival, their codes were 1 or 0, respectively. Then, the Prism program presented survival probability graphs.

### 2.15. Preparation of Rat Liver by Insulin Treatment

Female Rats (body weight approximately 300 g of Sprague Dawley) were administered with 1 mL insulin (10 U/kg body weight) in 0.9% saline through an intraperitoneal cavity injection of rat. Control groups were conferred with only 1 mL 0.9% saline in three clusters of rats. After 2 h insulin stimulation, rats were sacrificed, and livers were isolated from three groups. This study was conducted in accordance with the strict guidelines of the institutional Animal Studies Care and Use Committee of the Hallym University in Chuncheon, Korea (protocol no. Hallym 2017-3). Animal sacrifice was carried out using isoflurane anesthesia, and we attempted to reduce minimal pain and distress.

### 2.16. Xenograft Experiment

For the murine tumor model, BALB/c (female, 4–6 weeks old) and C57/BL-6J (female 4–6 weeks old, 15–17 g) mice were obtained from Samtako (Osan, Korea). The animal experiment protocol in this study was reviewed and approved by the Hallym University-Institutional Animal Care and Use Committee (IACUC) (Hallym 2021-6). Another protocol was similar to the previous report [16]. We did not use any chow model mice for tumor growth, and they were not fasted. For tumor implantation, non-transfected and transfected 4T1 cells were washed twice with serum-free culture solution, and 1 × 10^7^ cells (100 µL in PBS) were subcutaneously injected into the right flank of each mouse. When tumor volumes reached 100 mm^3^, tumor size measures were made three times a week using Vernier calipers. Volume (mm^3^) of tumors was calculated using the standard formula of length × width^2^, and growth curves were drawn by the Prism software (GraphPad). For tumor induction, we applied daily phorbol 12-myristate 13-acetate (PMA) solution (10 μL of 1 μM) near the tumor implanted area. The engrafted tumors were monitored for 28 days in the case of 4T1 cells, but for Hepa1c1c7 cells, only 20 days, and mice were then sacrificed, and their tumor samples were harvested. Mice were killed by urethane (widely used as an anesthetic) solution injection into the abdominal cavity to reduce minimal pain and distress.

### 2.17. Statistical Analysis

Statistical analysis and graphical presentation of all data were performed using GraphPad Prism 4 (GraphPad). In general, the data are shown as mean value ± SE of at least three independent experiments unless otherwise noted. The protein bands shown are representative of at least three independent experiments. Statistical analysis of significance (*p*-values) was based on one or two-way ANOVA or Student’s *t*-test in which *p* < 0.05 was considered significant and *p* < 0.001 as highly significant.

## 3. Results

### 3.1. p-PDHA1 Interacts with PKM2

Insulin was reported to increase p-Ser293 PDHA1 (p-PDHA1) levels in HepG2 and Huh7 hepatocellular carcinoma cells [14]. To elucidate a novel function of p-PDHA1 in HepG2 hepatocellular carcinoma cells, which are sensitive to insulin signaling, we identified the binding protein that may interact with p-PDHA1 using a recombinant GST phospho-mimic mutant PDHA1 Ser293Glu (S293D). Consequently, through MALDI-TOF analysis, PKM2 was identified to bind with p-PDHA1 (Figure 1A and Appendix A). Indeed, GST-PDHA1 S293D, a phosphomimetic form, strongly interacted with PKM2 in the lysates of HepG2 cells, which were stimulated with insulin, but not with the PKM2 in control cell lysates (Figure 1B). This suggests that additional factor(s) stimulated by insulin may be involved in p-PDHA1 and PKM2 interaction, for instance, additional post-translational modification(s) of either PDHA1 or PKM2. Before studying the detailed interaction between PKM2 and p-PDHA1, we assessed the levels of p-PDHA1 and PKM2 in several cell lines. Liver cell lines, Huh7, Hepa1c1c7, and HepG2, as well as WISH and HEK293 cells, revealed considerable levels of PKM2 and p-PDHA1 (Figure 1C). Endogenous expression levels of PKM2 and p-PDHA1 in each cell were different depending on the cell types, and they did not correlate with each other (Figure 1D). Insulin enhanced p-AKT level in HepG2 cells, suggesting insulin stimulates HepG2 cells in the classical pathway (Figure 1E) [14]. Of note, insulin augmented p-PDHA1 and PKM2 levels in HepG2 and Hepa1c1c7 (Figure 1F,G).

Confirmatively, p-PDHA1 co-immunoprecipitated with PKM2 in HepG2 and Hepa1c1c7 cells upon insulin treatment (Figure 1F,G, respectively). In addition, PKM2 co-immunoprecipitated with p-PDHA1 but neither the unphosphorylated PDHA1 nor PDC E2 and E3 subunits in HepG2 cells upon insulin treatment (Figure 1H). The alternatively different region between PKM1 and PKM2 corresponds to amino acids 379–435 (Figure 1J). In contrast to HepG2 cells, insulin attenuated p-PDHA1 levels in normal rat liver. Further, PKM1 instead of PKM2 was expressed in normal rat liver, but PKM1 did not co-immunoprecipitate with p-PDHA1 (Figure 1I). This suggested that p-PDHA1, particularly the domain around p-Ser293 residue of PDHA1, interacts with the 379–435 domain of PKM2 but not with that of PKM1. PKM2 consists of several domains, including the N-terminal domain (NTD), the C-terminal domain (CTD), A1, B, and A2 domains. Among them, the CTD of PKM2 mainly binds to p-PDHA1 (Figure 1J and Appendix A). Of note, overexpressed CTD of PKM2 disrupted co-immunoprecipitation of p-PDHA1/PKM2, suggesting that CTD is most likely involved in the p-PDHA1/PKM2 complex formation (Figure 1K). CTD of PKM2 abrogated cell proliferation (Figure 1L) and expression of c-Myc and cyclin D1 (Figure 1M). Instead of insulin, LPS treatment of the cells also enhanced the co-immunoprecipitation of p-PDHA1 and PKM2, suggesting that several stimulants may allow p-PDHA1/PKM2 complex formation (Figure 1N).

### 3.2. PKM2 Delivers p-PDHA1 to the Nucleus in Response to Insulin

p-PDHA1 was localized mainly to the mitochondria (data not shown). However, long-term administration (24 h) of insulin-induced nuclear localization of p-PDHA1 (Figure 2A). PKM2 moved comparatively more quickly, and both p-PDHA1 and PKM2 resided in the nucleus after 24 h of treatment with insulin (Figure 2B). In addition, we confirmed again by Western blotting that insulin augmented p-PDHA1 and PKM2 levels both in the nuclear and cytosolic functions (Figure 2C). In particular, si-PKM2 also markedly prevented the nuclear localization of p-PDHA1 post insulin treatment of the cells, suggesting that PKM2 facilitates nuclear localization of p-PDHA1 (Figure 2D). However, si-PDHA1 only slightly reduced the nuclear localization of PKM2 following treatment with insulin; these results suggested that p-PDHA1-PKM2 interaction facilitates nuclear translocation of both proteins (Figure 2E).

### 3.3. Target Genes of 1-PDHA1 and PKM2

Chromatin immunoprecipitation sequencing (ChIP-Seq) by using p-PDHA1 and PKM2 antibodies in HepG2 cells upon insulin allowed us to identify common genes, the promoters of which were associated with p-PDHA1 and PKM2 (Figure 3A). The mRNA levels of target genes, including KDM1B, GPR174, and long intergenic non-protein coding RNA 00273 (LINC00273), were also increased (Figure 3B). As we found that LINC00273 was involved in EMT, which is critical for tumor cell migration, through ZEB1 expression, we focused on the LINC00273 gene expression as a representative of common genes that were regulated by the p-PDHA1/PKM2 complex in transcriptional level [20,21]. First, we confirmed that si-LINC00273 significantly abolished LINC00273 expression by using the RT-PCR experiment (Figure 3C). Furthermore, si-LINC00273 significantly attenuated cell proliferation, suggesting LINC00273 is required for cell proliferation upon insulin (Figure 3D). p-PDHA1 and PKM2 were found to indeed associate with the promoter of the LINC00273 gene following ChIP-PCR (Figure 3E). Post insulin treatment of the cells, si-PDHA1 and PDHA1 S293A (dephospho-mimic) reconstitution also effectively abolished LINC00273 expression while addback with PDHA1 WT and S293D, but not S293A restored LINC00273 expression, suggesting the phosphorylated Ser293 form of PDHA1 is crucially required for LINC00273 expression (Figure 3F). Likewise, RT-qPCR also demonstrated that si-PDHA1 and PDHA1 S293A dephospho-mimic form prevented LINC00273 mRNA expression while reconstituted RPHA1 WT and S293D phosphomimetic recovered its expression (Figure 3H). Additionally, si-PKM2 also attenuated LINC00273 levels while reconstituted PKM2 restored its levels post insulin treatment, suggesting that PKM2, as well as p-PDHA1, is critically required for LINC00273 expression (Figure 3G). RT-qPCR also clarified that si-PKM2 also attenuated LINC00273 levels while reconstituted PKM2 restored its levels (Figure 3I).

### 3.4. p-PDHA1 and PKM2 Induces Cell Migration through Expression of LINC00273 and ZEB1

In the presence of insulin, si-LINC00273 treatment of cells attenuated proteins expression of EMT markers, including ZEB1, N-cadherin, vimentin, and Snail1, while it augmented E-cadherin levels (Figure 4A). Similarly, si-PKM2 also attenuated ZEB1, N-cadherin, and vimentin protein levels but increased E-cadherin expression in the presence of insulin, suggesting that PKM2 is involved in the regulation of these protein expressions, possibly through LINC00273 (Figure 4B). si-PDHA1 treatment of cells also reduced ZEB1 expression; this was able to be compensated in the addback of PDHA1 (WT and S293D), restoring ZEB1 expression in the presence of insulin. However, PDHA1 S293A, the dephospho-mimic, was not able to restore ZEB1 expression, suggesting that p-PDHA1, as well as PKM2, is critical for ZEB1 level increase upon insulin treatment (Figure 4C).

si-PDHA1 reduced cell migration in the wounding assay, but the addback of WT PDHA1 or PDHA1 S293D restored migration in response to insulin. In contrast, addback with PDHA1 S293A still had reduced cell migration in response to insulin, suggesting that p-PDHA1 is required for cell migration, likely through the role of required EMT proteins (Figure 4D and Appendix A). si-PKM2 also reduced cell migration in the same assay and reconstituted PKM2 restored that (Figure 4E and Appendix A). Similarly, si-LINC00273 allowed a decrease in cell migration in response to insulin (Figure 4F and Appendix A). Interestingly, primary tumor samples have more expressed ZEB1 compared to normal tissues with data obtained from ULCAN, cancer omics data site (http://ulcan.path.uab.edu/ accessed on 1 January 2021). (Figure 4G). The higher levels of ZEB1 message in patients harboring liver hepatocellular carcinoma also correlate with poorer survival with analysis using the Kaplan–Meier plotter site (https://kmplot.com/analysis/ accessed on 10 February 2021) (Figure 4H).

### 3.5. p-PDHA1/PKM2 Complex Increases Histone Acetylation in Response to Insulin

As PKM2 has been reported to interact with various transcription factors [8], we explored whether p-PDHA1 also interacted with the transcription factors. STAT3, HIF-1α, and Oct4 co-immunoprecipitated with both PKM2 and p-PDHA1 (Figure 5A–C). Indeed, insulin-induced histone acetylation was detected with acetylation-specific antibodies, including Ac-H3K27, Ac-H3K9/K14, and Ac-H3K18 in HepG2 cells (Figure 5D). si-PDHA1 and PDHA1 S293A dephospho-mimic form presenting a high enzyme activity had dramatically lower histone acetylation, including for Ac-H3K27, Ac-H3K9/K14, and Ac-H3K18. On the other hand, PDHA1 WT and PDHA1 S293D phospho-mimic form presenting low enzyme activity maintained their increases in histone acetylation upon insulin activation; this suggested that the p-Ser293 form of PDHA1, not due to enzyme activity but due to another mechanism, is critically required for histone acetylation increases upon insulin treatment (Figure 5E). Remarkably, p-PDHA1 co-immunoprecipitated with p300 histone acetyltransferase (HAT) and ATP citrate lyase (ACL) catalyzes the conversion of citrate/CoA to acetyl-CoA/oxaloacetate (Figure 5F). In turn, p300 HAT co-immunoprecipitated p-PDHA1 as well as PKM2 (Figure 5G). These results strongly suggested that p-PDHA1 could recruit p300 HAT and regulate histone acetylation in response to insulin. Indeed, si-p300 HAT and si-ACL treatment of cells prevented transcriptional increases in expression of LINC00273 and KDM1B, suggesting that LINC00273 and KDM1B expression are actually regulated by histone acetylation (Figure 5H). Furthermore, insulin enhanced the acetylation of H3K9/14, H3K18, and H3K27 in the promoter region of the LINC00273 gene (Figure 5I). Indeed, we confirmed again that H3K27 was acetylated in the promoter region of LINC00273 at the site of the UCSC genome browser (Human GRCh37/hg19, chromosome 16: 33960925-33961400). The region was localized near the identified region (chr16:33961052-33962503) (Figure 5J).

### 3.6. p-PDHA1 Is Relevant to Tumorigenesis in Liver

At the cellular level, si-PDHA1 and reconstituted PDHA1 S293A reduced cell proliferation while reconstituted PDHA1 WT and S293D recovered it in 4T1 cells (Figure 6A). Regarding p-PDHA1 in tumorigenesis, si-PDHA1 and reconstituted PDHA1 S293A dephospho-mimetic while reconstituted PDHA1 WT and S293D restored tumor growth with 4T1 cells in xenograft experiments (Figure 6B). The 4T1 cells expressing si-PDHA1 and co-expressing PDHA1 S293A dephospho-mimetic, but not PDHA1 WT or PDHA1 S293D phospho-mimetic, had reduced xenograft tumor growth, as detected by tumor volume changes (Figure 6B–D). The terminal total mice body weights were not significantly changed in any of the xenograft groups (data not shown). We then analyzed p-PDHA1 levels by using Western blotting from human liver cancer patients (Figure 6E). Then, we assessed the survival probability of liver cancer patients with high versus low expressing p-PDHA1. Contrary to our expectation, p-PDHA1 did not display differential survival for the liver cancer patients analyzed, likely due to alteration of endogenous insulin concentration in patients (Figure 6F,G and Appendix A). In addition, we determined PKM2 and p-Tyr105 PKM2 levels by Western blotting in a specimen of liver cancer patients (Figure 6E). The higher levels of PKM2 and p-Tyr105 PKM2 expression in the same liver cancer patients gave rise to poorer survival probability (Figure 6I,K). In addition, the lung cancer patients with the higher p-PDHA1 and PKM2 levels, when determined by Western blotting of tumor samples (Figure 6L,M,O), markedly reduced survival probability (Figure 6N–P and Appendix A). We could not distinguish significant differences between stages, likely due to the small number of patients.

### 3.7. Proposed Schematic Illustration for Role of p-PDHA1/PKM2 Complex in Cancer Cells

Insulin induces Ser264 phosphorylation of PDHA1. The p-PDHA1/PKM2 complex translocate into the nucleus, where the complex regulates histone acetylation by recruiting p300 HAT and ACL to the promoter of specific genes such as LINC00273, resulting in LINC00273 expression. miR200 interferes with ZEB1 expression through binding to ZEB1 mRNA. However, expressed LINC00273 captures miRNA200, and ZEB1 mRNA can be used for ZEB1 protein expression. Conclusively, insulin increases cancer cell proliferation/migration and tumorigenesis of hepatocellular carcinoma cells.

## 4. Discussion

### 4.1. Functions of p-PDHA1/PKM2 Complex in the Nucleus

Here, we first proposed that p-PDHA1 binds to PKM2 and regulates gene expression. The structure of PDHA1 phosphorylated at Ser293 residue may be different from the dephospho-form. Herein, p-Ser293 of PDHA1 may facilitate its binding to CTD of PKM2 (Figure 1J). Moreover, it was novel that PKM2 is able to drive or deliver p-PDHA1 to the nucleus through p-PDHA1/PKM2 complex, although HSP70 was previously reported to deliver PDC from mitochondria to the nucleus upon EGF stimulation [22]. PKM2 is bound to p-PDHA1 but not to PDHE2 nor PDHE3 subunit in the nucleus. As the PKM2 in the nucleus was reported to be monomeric [23], we speculated that PKM2 binds to p-PDHA1 in monomeric form. PDC, the PDH complex consisting of E1, E2, and E3 subunits, was reported to be involved in histone acetylation, positively mediated via acetyl-CoA production in the nucleus [22]. Thus, we predicted first that as PDHA1 phosphorylation at Ser293 has reduced its enzymatic activity in producing acetyl-CoA, this would lead to reduced histone acetylation once p-PDHA1 forms PDC in the nucleus. Contrary to our prediction, insulin markedly enhanced histone acetylation, including increased levels of Ac-H3K27, Ac-H3K9/K14, and Ac-H3K18 (Figure 5D), and PDHA1 S293A de-phospho-mimic form dramatically suppressed Ac-H3K27, Ac-H3K9/K14, and Ac-H3K18 levels in HepG2 cells. This suggested that Ser293 phosphorylation of PDHA1 is crucially required for histone acetylation (Figure 5E). p-PDHA1 regulates histone acetylation in specific target genes through the recruitment of p300 HAT and ACL.

p-PDHA1 and PKM2 were shown to bind with several transcription factors, including STAT3, HIF-1α, and Oct4 (Figure 5A–C). STAT3 has been reported to bind to PKM2 with PKM2 phosphorylating Tyr705 of STAT3 [13]. Nuclear PKM2 was reported to bind to Oct4 [24], and dichloroacetate, a PDH kinase inhibitor, which attenuates p-PDHA1 levels, increases the levels of PKM2-Oct4 complex and inhibits Oct4-dependent gene expression [25]. These results suggest several transcription factors may also be involved in p-PDHA1/-PKM2 functions, although the detailed mechanism remains to be discovered. However, among the transcription factors, HIF-1α has been reported to interact with p300 HAT [26,27]. In addition, PKM2 is a co-activator for HIF-1α, a transcription factor for response to hypoxia [28]. In our analysis, HIF-1α/p-PDHA1 interaction was markedly increased by insulin treatment (Figure 5B). Thus, we assumed that PKM2/p-PDHA1/HIF-1α/p300 HAT complex formation may be mainly induced by insulin stimulation for the histone acetylation.

### 4.2. LINC00273, a Target Gene That Is Regulated by p-PDHA1 and PKM2

From the ChIP-Seq data set, we focused on LINC00273 as a representative gene induced by insulin through p-PDHA1/PKM2 interaction. LINC00273 expression is elevated in tumor samples for various organs, and LINC00273 levels correlate with the degree of cellular de-differentiation, which is an important feature of tumor invasion [20]. Growth factor TGF-β stimulates metastasis by inducing ZEB1 and LINC00273 in A549 cells, and knockdown of LINC00273 reduces TGF-β-mediated ZEB1 expression as well as metastasis [21]. ZEB1 is one of the EMT-associated transcription factors [29] and is a transcriptional repressor of epithelial genes, including E-cadherin [30]. ZEB1 binds to the promoter of the E-cadherin gene, leading to repression of E-cadherin expression [31,32]. LINC00273 can bind to miR200a-3p in response to TGF-β, but otherwise, miRNA200a-3p binds to ZEB1 mRNA in the absence of TGF-β stimulation, leading to the destruction of ZEB1 mRNA [21]. We observed that LINC00273 was engaged in cell proliferation (Figure 3D) as well as cell migration in HepG2 cells (Figure 4F). However, the mechanism of LINC00273 for cell proliferation remains to be disclosed.

Not all the target genes of p-PDHA1 and PKM2 were not mainly studied here. However, insulin also enhanced the expression of GRP174, KDM1B (Figure 3B) through p-PDHA1 and PKM2., also referred to as lysine-specific demethylase 2 (LSD2). Actually, GPR174 [33,34], and KDM1B [35] play significant roles in tumorigenesis.

### 4.3. Elevated Levels of p-PDHA1 and PKM2 in Hepatocellular Carcinoma Cells

p-PDHA1 is increased in a variety of cancer types, and this is mediated by several types of PDH kinases (PDKs) overexpressed in numerous cancers [36]. p-PDHA1, which reduces PDH activity, is critical for increased lactate and other intermediate metabolites during glycolysis and is used for cancer cell growth, which is called aerobic glycolysis or the Warburg effect [37,38]. Thus, it is possible that insulin contributes to the Warburg effect by producing p-PDHA1 in HCC cells. Indeed, insulin was reported to increase lactate levels in HepG2 and Bel7402 HCC cells in vitro [39]. However, the high and low levels of p-PDHA1 did not distinguish differences in survival probability, contrary to our expectation (Figure 6G). This may be due to the different transient insulin concentrations in collected blood samples from liver cancer patients. Indeed, insulin sensitively augmented both PDHA1 and p-PDHA1 levels in hepatocellular carcinoma cell lines (Figure 1E–G). In the case of lung cancer, the patients harboring the higher p-PDHA1 and PKM2 levels revealed much poorer survivability (Figure 6L–P).

Notably, a large number of metabolic enzymes have noncanonical or nonmetabolic functions. Of note, several metabolic enzymes such as ketohexokinase isoform A (KHK-A), hexokinase (HK), nucleoside diphosphate kinase 1 and 2 (NDP kinase1/2: NDKP1/2), and phosphoglycerate kinase 1 (PGK1), as well as PKM2, are also able to function as protein kinases that phosphorylate a variety of protein substrates to regulate the Warburg effect, gene expression, cell cycle progression and proliferation and many other underlying cellular functions [40,41]. For instance, PGK1 localized in cytosol, nucleus, or mitochondria regulates specific cellular functions. Mitochondrial PGK1 plays a role as a protein kinase, which phosphorylates pyruvate dehydrogenase kinase 1 (PDHK1) at Thr338, thereby leading to PDHK1 activation. Then the active PDHK1 phosphorylates PDHA1 at Ser293, ensuring PDH inactivation, lactate production, and brain tumorigenesis [42]. Although insulin has been well established to regulate gene expression through other pathways such as the Ras signaling pathway, we proposed in this study a novel mechanism by which insulin regulates gene expression through p-PDHA1 along with PKM2 in hepatocellular carcinoma HepG2 cells. Of interest, hyperinsulinemia and insulin resistance may be associated with an increased risk of hepatocellular carcinoma (HCC) [43]. DM2 was reported to be associated with a three-fold increased risk of heparocarcinoma occurrence [44]. Hereby, it was necessary to study the effect of insulin on HCC cells. However, we have the limitation that we did not use any diabetic/hepatocellular carcinoma clinical patient samples in this study. Therefore, it remains to be elucidated in a further clinical study that p-PDHA and PKM2 complex in the HCC patients is crucial for the HCC progress. 

## 5. Conclusions

In summary, we proposed a novel underlying mechanism that insulin induces an increase in p-PDHA1 levels and allows p-PDHA1 to translocate into the nucleus along with PKM2. The p-PDHA1/PKM2 complex regulates histone acetylation of LINC00273 promoter through recruitment of ACL and p300 HAT (Figure 7).

## Figures and Tables

**Figure 1 biomedicines-10-01256-f001:**
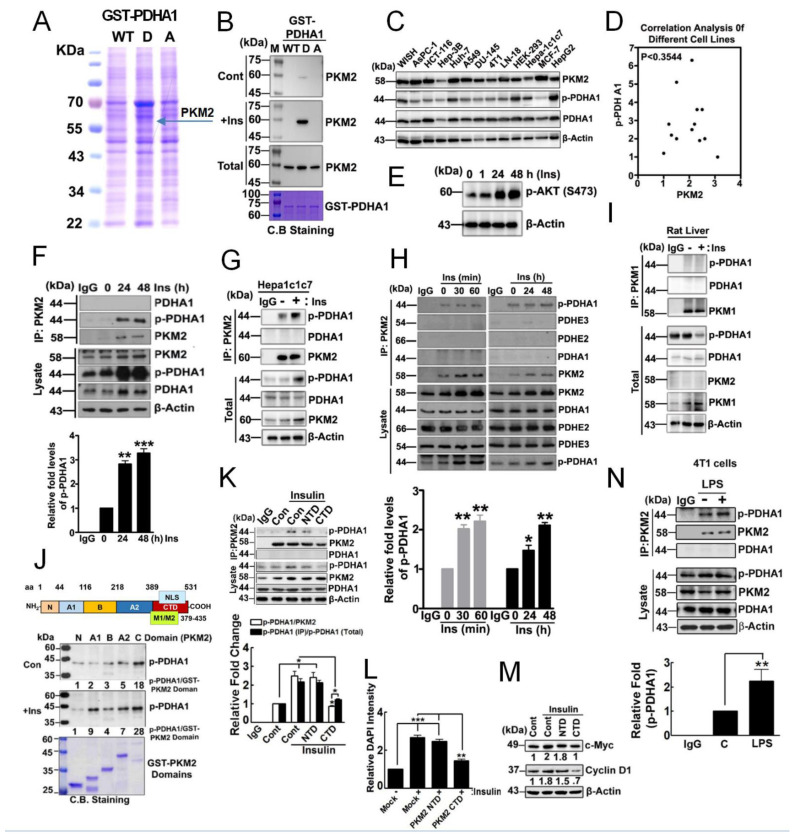
**p**−**Ser****293****PDHA1 binds to PKM2.** (**A**) PC12 cells were stimulated with NGF100 ng/mL, lysate was then incubated with purified GST-PDHA1 WT, S293D, and S293A mutant beads, proteins were separated by SDS-PAGE, and target protein species identified through MALDI-TOF analysis. (**B**) Purified recombinant protein beads were incubated with HepG2 cell lysates treated with insulin 100 nM for 48 h, and bound PKM2 was identified by immunoblotting. (**C**) Relative PKM2, PDHA1, and p-PDHA1 levels in several cancer cell lines were determined by immunoblotting. (**D**) Correlation between p-PDHA1 and PKM2 levels in several cancer cell lines is presented. (**E**) p-Ser473 AKT levels in HepG2 cells were detected by Western blotting in the presence of insulin (100 nM) in indicated time. (**F**–**H**) HepG2 and Hepa1c1c7 cells were stimulated with insulin (100 nM in indicated time). Co-immunoprecipitation with PKM2 and then indicated proteins were detected. (**I**) Insulin (10 U/kg body weight for 2 h) treated in rat liver was prepared and immunoprecipitated with PKM1, and interaction measured by p-PDHA1 antibody. (**J**) Insulin (100 nm for 48 h) or without insulin cells lysate were incubated with each recombinant GST-PKM2 domain (2 µg protein), then indicated proteins were detected (upper panel). GST-PKM2 domains were visualized by Coomassie blue staining (lower panel). (**K**) CTD, NTD domains of PKM2 were overexpressed in HepG2 cells and PKM2/p-PDHA1 complex formation measured by Western blotting in the presence of insulin (100 nM for 24 h) (**L**) HepG2 cells transfected with CTD and NTD domains of PKM2 and then cell proliferation measured by DAPI staining with or without insulin (100 nM for 24 h). (**M**) HepG2 cells tranfected with CTD and NTD of PKM2 and stimulated with or without insulin (100 nM for 24 h), and c-Myc, Cyclin D1 levels were measured by Western blotting. (**N**) 4T1 cells were treated with LPS (10 μg/mL at 4 h); PKM2 was immunoprecipitated, and then indicated proteins were detected. * *p* < 0.05, ** *p* ˂ 0.01; *** *p* ˂ 0.001.

**Figure 2 biomedicines-10-01256-f002:**
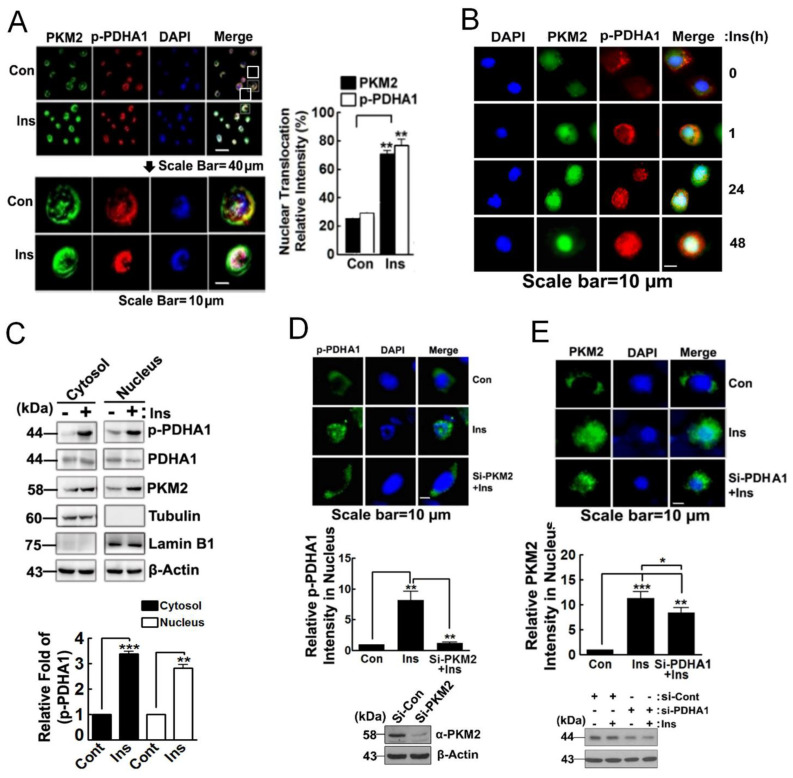
**PKM2****delivers****phosphorylated-Ser293 PDHA1 (p**−**PDHA1) to the nucleus.** (**A**,**B**) HepG2 cells were stimulated with 100 nM insulin for 48 h (**A**) or 1, 24, and 48 h (**B**), p-PDHA1 and PKM2 changes were identified with immunohistochemistry. (**C**) HepG2 cells were treated with 100 nM insulin for 48 h, and cytosolic and nuclear fractions were separated, and then p-PDHA1 and PKM2 were determined by Western blotting. (**D**) si-PKM2 was transfected into HepG2 cells and insulin (100 nM for 48 h) was added to the cells, and p-PDHA1 levels were visualized with immunohistochemistry. Knockdown PKM2 levels were measured by Western blotting. (**E**) si-PDHA1 was transfected, then PKM2 was visualized with immunohistochemistry followed by insulin activation (100 nM for 48 h). PDHA1 silencing level was confirmed by immunoblotting. Statistic data were showed as * *p* < 0.05, ** *p* ˂ 0.01; *** *p* ˂ 0.001 and scale bar was 10 µm.

**Figure 3 biomedicines-10-01256-f003:**
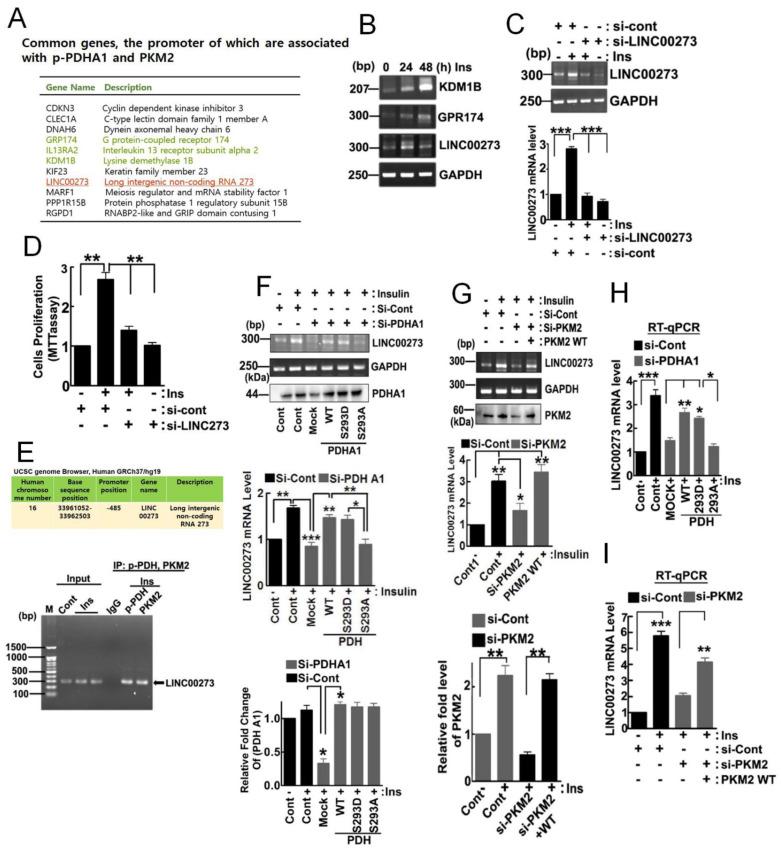
**p**−**PDHA1 binds to the promoter of specific genes and regulates their expression.** (**A**) Promoters of genes that were associated with p-PDHA1 and PKM2 were identified by ChIP sequencing. (**B**) HepG2 cells were treated with insulin (100 nM in indicated time) and mRNA levels of KDM1B, GPR174, and LINC00273 were determined by RT-PCR. (**C**) HepG2 cells were transfected with si-control and si-LINC00273 and then stimulated by 100 nM insulin for 24 h. Then, silencing of LINC00273 mRNA level measured by RT-PCR. (**D**) HepG2 cells were transfected with si-LINC00273 and stimulated with or without 100 nM insulin for 24 h. Then cell proliferation was measured by the MTT assay. (**E**) ChIP-PCR was conducted with primers of LINC00273 from the HepG2 cells stimulated with insulin (100 nM for 48 h). (**F**) HepG2 cells were transfected with si-PDHA1 with the addback PDHA1 WT, S293D, and S293A mutants with Insulin (100 nM for 24 h) and LINC00273 mRNA expression was measured via RT-PCR from the harvested RNA. (**G**) HepG2 cells were transfected with si-PKM2 and then reconstituted with a PKM2 WT expression vector. Insulin (100 nM for 24 h) was added to the cells, and LINC00273 mRNA expression was measured via RT-PCR from the cDNA. (**H**) HepG2 cells were transfected with si-PDHA1 and reconstituted with PDHA1 mutants (S293D and S293A) in the presence of insulin or without insulin (100 nM for 24 h), then LINC00273 mRNA expression was measured by RT-qPCR (**I**) HepG2 cells were knockdown with si-PKM2 and addback with PKM2-WT and mRNA level of LINC00273 was quantified through RT-qPCR. (* *p* < 0.05, ** *p* ˂ 0.01; *** *p* ˂ 0.001).

**Figure 4 biomedicines-10-01256-f004:**
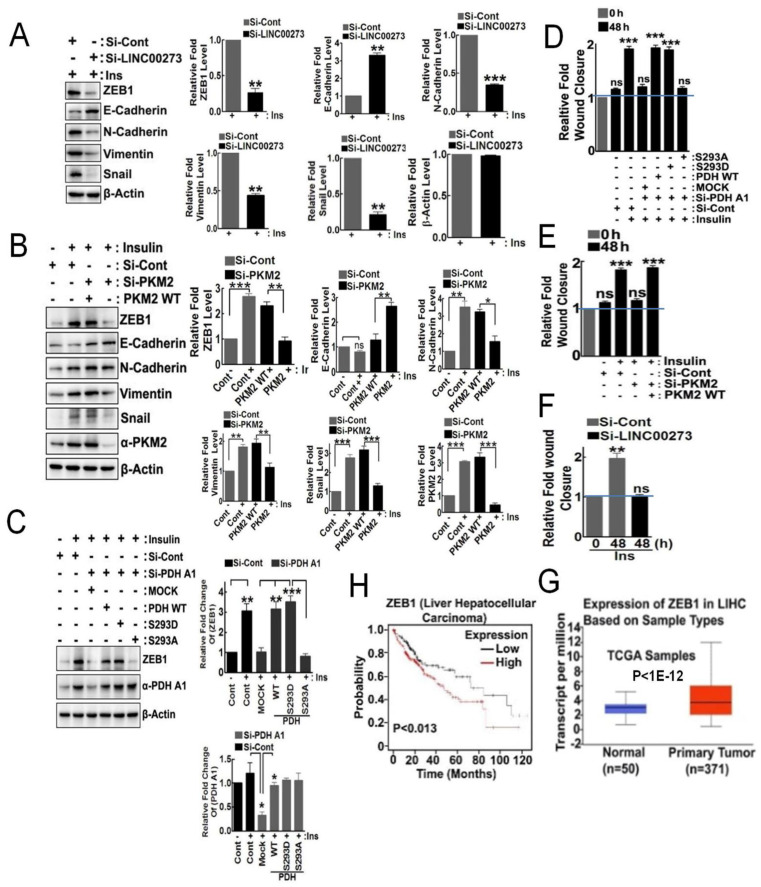
**LINC00273 regulates ZEB1 expression and cell migration**. (**A**) HepG2 cells were transfected with si-LINC00273 upon 100 nM insulin (100 nM for 48 h), and expression changes of ZEB1, E-cadherin, N-cadherin, vimentin, and Snail were determined by immunoblotting and plotted in bar diagram. (**B**) HepG2 cells were si-PKM2 transfected with PKM2 WT addback and expression and stimulated with insulin (100 nM for 48 h), and modulations of ZEB1, E-cadherin, N-cadherin, vimentin, and Snail were determined by immunoblotting and represented as bar diagram. (**C**) si-PDHA1 was transfected into HepG2 cells, and PDHA1 WT, S293D, or S293A was addback to the cells, which were stimulated with 100 nM insulin (100 nM for 24 h), and the expression changes for ZEB1 and PDHA1 were determined by immunoblotting and quantification was shown in bar diagram. (**D**) si-PDHA1 transfection and addback with PDHA1 WT, S293D, and S293A to the HepG2 cells, which were stimulated with insulin (100 nM for 48 h), and cell migration was measured by the wound healing assay, and the relative wound closure shown in bar diagram. (**E**) HepG2 cells were transfected with si-PKM2 and in the presence of PKM2 WT addback. Cell migration upon insulin (100 nM for 48 h) was measured by wound healing assay and presented as bar diagram. (**F**) HepG2 cells were transfected with si-LINC00273 and cell migration was determined by the wound healing assay followed by insulin (100 nM for 48 h) and wound levels were analyzed by bar diagram. (**G**) ZEB1 expression was compared between normal tissues and tumor tissues of liver hepatocellular carcinoma (HCC) (http://ulcan.path.uab.edu/ accessed on 1 January 2021). (**H**) Survival probabilities for low and high ZEB1 expression in several cancer types were obtained and plotted via Kaplan–Meier plotter (Pan-cancer RNAseq, TCGA survival). (* *p* < 0.05; ** *p* ˂ 0.01; *** *p* ˂ 0.001).

**Figure 5 biomedicines-10-01256-f005:**
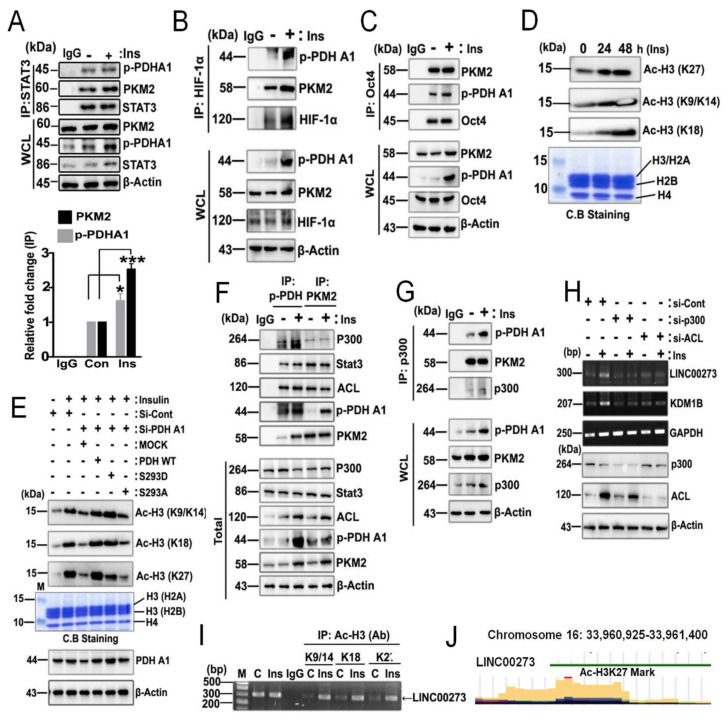
**Regulation of Histone acetylation through p**−**PDHA1/PKM2 Complex upon insulin stimulation.** (**A**–**C**) HepG2 cells were stimulated with Insulin (100 nM at 24 h), and STAT3, HIF-1α, and Oct4 were immunoprecipitated, and p-PDHA1 and PKM2 were detected by Western blotting. (**D**) HepG2 cells were treated with insulin (100 nM) in indicated time, and acetylated histones (Ac-H3K27, Ac-H3K9/14, and Ac-H3K18) levels were measured from extracted histone proteins by Western blotting. Histone proteins were visualized by Coomassie blue staining in SDS-PAGE (lower panel). (**E**) HepG2 cells were knockdown with si-PDHA1, then addback with PDH WT, 293D, and 293A mutant, and then stimulated with insulin (100 nM at 24 h), and indicated proteins were detected. (**F**) Insulin (100 nM at 24 h) stimulated HepG2 cells lysate were immunoprecipitated with PKM2, p-PDHA1, and then p300, STAT3, and ACL levels were detected. (**G**) HepG2 cells stimulated by insulin (100 nM at 24 h), cell lysates were used for immunoprecipitation with p300 HAT antibody, and then co-immunoprecipitated p-PDHA1 and PKM2 were determined by Western blotting. (**H**) LINC00273 and KDM1B mRNA levels were measured through RT-PCR from HepG2 cells, which were knockdown with si-ACL and si-p300 and stimulated with or without insulin (100 nM at 24 h) treatment. (**I**) ChIP-PCR of LINC00273 genes was performed after immunoprecipitation of several acetylated histone proteins by using Ac-H3K9/14, Ac-H3K18, and Ac-H3K27 antibodies in the presence of insulin (100 nM at 24 h). (**J**) Acetylated H3K27 in the promoter region of LINC00273 was derived from the site of the UCSC genome browser (genome.ussc.edu, Human GRCh37/hg19, chromosome 16: 33960925-33961400). (* *p* < 0.05; *** *p* ˂ 0.001).

**Figure 6 biomedicines-10-01256-f006:**
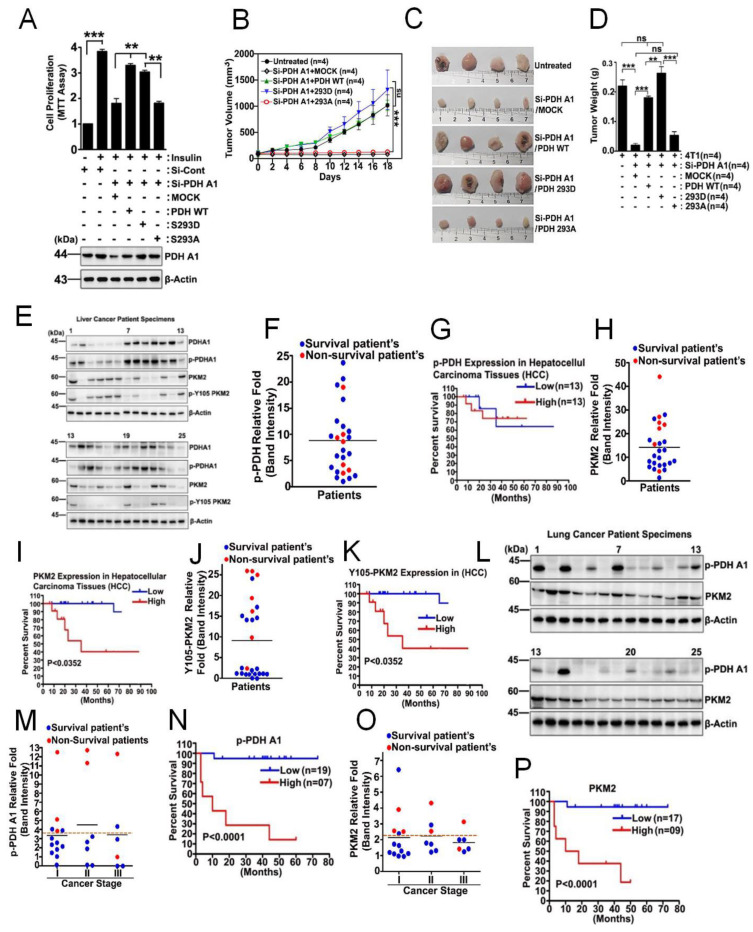
**Increased p**−**PDHA1 levels correlate with poor survival probability.** (**A**) 4T1 cells were treated with si-PDHA1, then addback with PDHA1 WT, S293D, or S293A, then activated with insulin (100 nM) for 48 h. Cell proliferation was assessed with an MTT assay, and the PDHA1 levels were determined by Western blotting. (**B**,**C**) Tumor volumes and tumor mass of 4T1 xenografts by using BALB/c mice containing si-PDHA1, plus the addback vector PDHA1 WT, S293D, or S293A mutants, were measured twice a week. (**D**) Tumor weight of 4T1 xenografts containing si-PDHA1 and the addback PDHA1 WT, S293D, or S293A mutants were measured. (**E**) p-PDHA1, PDHA1, PKM2, and p-Tyr105 PKM2 levels were determined with Western blotting from human liver cancer tissue samples. (**F**,**H**,**J**) Relative levels of p-PDHA1, PKM2, and p-Tyr105 PKM2 for human liver tumor samples were shown, and the survival/non-survival status of patients are noted in blue and red, respectively. (**G**,**I**,**K**) Survival probability according to the relatively low and high levels of p-PDHA1, PKM2, and p-Tyr105 PKM2 in human liver cancer patients are presented in a Kaplan–Meier graph by using GraphPad Prism 4 version. (**L**) p-PDHA1 and PKM2 levels in human lung cancer patient samples were measured by Western blotting. (**M**,**O**) Relative band intensities of p-PDHA1 and PKM2 levels in lung cancer patient samples were shown depending on the stage, and the survival/no-survival status of the patient is noted in blue and red, respectively. (**N**,**P**) Survival probability according to the tumor-associated relatively low and high levels of p-PDHA1 and PKM2 in lung cancer patients is presented in Kaplan–Meier graph. (** *p* ˂ 0.01; *** *p* ˂ 0.001).

**Figure 7 biomedicines-10-01256-f007:**
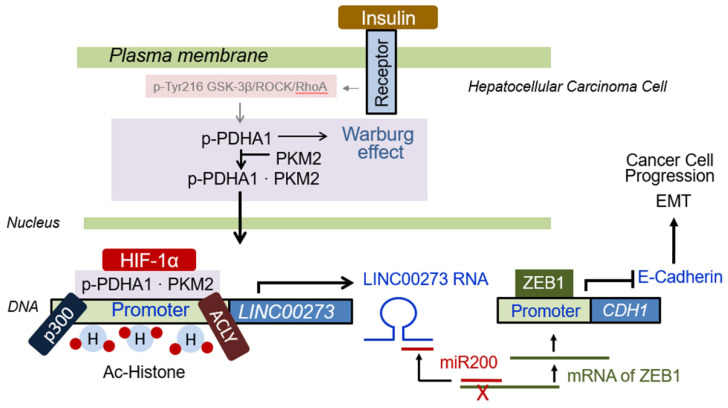
**Proposed schematic for the role of p**−**PDHA1/PKM2 complex in cancer cells.** Insulin induces Ser264 phosphorylation of PDHA1. Then p-PDHA1/PKM2 complex translocate into the nucleus, where the complex regulates histone acetylation by recruiting p300 HAT and ACL to the promoter of specific genes such as LINC00273, resulting in LINC00273 expression. miR200 interferes with ZEB1 expression through binding to ZEB1 mRNA. However, expressed LINC00273 captures miRNA200, and ZEB1 mRNA can be used for ZEB1 protein expression. Conclusively, insulin increases cancer cell proliferation/migration and tumorigenesis of hepatocellular carcinoma cells.

## Data Availability

Data is contained within the article or Appendix A.

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
