# Peer review of "Pyruvate Dehydrogenase A1 Phosphorylated by Insulin Associates with Pyruvate Kinase M2 and Induces LINC00273 through Histone Acetylation"

_biomedicines, 2022, doi:10.3390/biomedicines10061256_

Round 1
Reviewer 1 Report
Hossain et al. provide an interesting study concerning novel roles of pyruvate dehydrogenase (PDHA1) and pyruvate kinase (PKM2). A major finding is that PDHA1, PKM2, and HIF-1α interact and are involved in regulation of LINC00273. This study certainly provides an interesting new insight into interaction between metabolic enzymes, HIF-1α, and histone acetylation. The authors performed an extensive amount of work; indeed, one of the major strengths of the study is the amount and versatility of the results, which range from cell cultures, including gene silencing and overexpression, to animal studies as well as assessment of clinicals samples and data.
Comments/questions:
Introduction:
- “Growth hormone (GH) such as insulin…” GH and insulin are two different hormones. The sentence should be revised.
- “Of interest, diabetes mellitus (DM) patients induced by insulin resistance and hyperinsulinemia…” This sentence should be revised so that it will describe the pathogenesis of (type 2) diabetes more precisely.
- Regulation of pyruvate dehydrogenase by metabolites should be at least mentioned so that it does not come across that phosphorylation is the only mechanism.
Methods:
- General comment: to fully appreciate the data provided methods should be described in more detail. Please see specific comment below.
- Cell culturing should be described more precisely: For instance, what was the glucose concentration in media? Were antibiotics/antimycotics and serum present in the media during experiments?
- Gene silencing should be described in more detail. For instance, 1) what was the concentration of siRNA, 2) at what point after addition of siRNA were experiments performed, 3) was there only one siRNA or was there a pool of siRNAs, etc.
- Conditions under which PCR was performed should be described in more detail.
- Several animal models were used in the study: which ethics approval refers to the insulin treatment in rats (paragraph 2.16)?
- The method by which experimental animals were killed should be reported. Also, what kind of chow did they have, were they fasted or not? If yes, for how long, etc.
- Quantification of immunocytochemistry data should be described in more detail.
- Description of clinical characteristics of patients included in the study is insufficient.
- Xenograft experiment should be described in more detail.
- Methodology of measuring tumours in mice in vivo should also be described.
Results:
- General comments:
- It would be better if immunoblots were quantified and statistically analysed wherever possible. This should be done at least for the most important results.
- The presence of serum may affect the results of insulin treatment. At least for a few critical experiments, the authors should show that the classical insulin pathway was activated (e.g. Akt) by insulin.
- Fig. 1B: what was the insulin concentration and how long was the treatment? Insulin appears to have stimulated interaction with the phosphomimetic mutant, but not control cells. Could authors comment this result?
- Fig. 1I: What was the insulin concentration and how long was the treatment?
- Fig. 1M: For how long were cells treated with LPS?
- As mentioned above, quantification (densitometry) should be performed for data in Fig. 1. Also, the total PDHA1 should be included as a control in all blots where p-PDHA1 was assessed.
- Fig. 2: Scale bars are missing from lower magnification micrographs.
- Fig. 2C requires a control blot – total PDHA1.
- Fig. 2: How effective was gene silencing of PKM2 and PDHA1? A control experiment showing the effectiveness of gene silencing of PKM2 and PDHA1 should be shown (at least immunoblot).
- Fig. 2: MTT assay is an indirect measure of cell proliferation: i.e. it does not assess cell proliferation directly. How was cell proliferation assay performed (how many cells were seeded, what was media composition – serum, antibiotics, antimycotics)?
- Fig. 3: It would be important to assess proliferation (MTT assay) of cell cultured treated with si-LINC00273 in the absence of insulin. This is an important control experiment.
- Fig. 3: How effective was silencing of LINC00273? It would be important to show that its knock-down was effective by performing PCR and showing quantified results. Similarly, silencing of PDHA1 and PKM2 should be assessed quantitatively. While Fig. 4 shows effective silencing in insulin-treated cells, at least a few controls should be shown for results in Fig. 2 and 3 (blots or immunocytochemistry, depending on the type of analysis).
- Fig. 2 and 3: Agarose gel electrophoresis was used to quantify RNA. However, a real-time PCR would provide a more precise (more quantitative) result.
- Fig. 4 and 6: Clinical correlations are interesting and relevant; however, their usefulness is limited if patients’ characteristics are not indicated. Were patients in Fig. 4G and H in the same cancer stage? Could the stage have affected ZEB expression and therefore confounded the results in Fig. 4H? Also, is there any difference between cancer patients in Fig. 6F, H, and J who survived and those who did not survive? Did those who died have a higher cancer stage etc.? Did phosphorylation differ between stages (e.g. was phosphorylation higher in those with a higher stage?) or did phosphorylation correlate with a poorer prognosis for patients with the same cancer stage? To what time after diagnosis does the “live/dead” status of patients refer to? Did patients die of cancer or did perhaps some die of other causes?
- Also, I suggest the authors to consider different terminology with regard to the patients – “live/dead status” is usually used for cell culture and may not be the most appropriate choice of words for patients.
- Fig. 5: The text and/or figure legend should always make absolutely clear which cells were used for experiments and what were the treatments (insulin concentration, duration of treatment etc.).
- Fig. 5A: There appears to be less p-PDHA1 upon insulin treatment (IP results)? What does the quantification and statistical analysis indicate?
- Fig. 6: siPDHA1 and siPKM2 cells were used for the xenograft experiment; however, how does silencing of PDHA1 and PKM2 affect the in vitro viability of Hepa1c1c7 and 4T1 cells? Were they capable of proliferating normally, was their survival reduced/apoptosis increased? These aspects require a more detailed assessment because they are important for interpretation of the in vivo results.
- Fig. 6: Were insulin concentrations measured in patients with HCC? If not, the reference to possible effect of different insulin concentrations should be omitted from the Results.
- HCC has different underlying causes: what kind of patients were included in the study: those with type 2 diabetes, HCV infection, alcoholic (or non-alcoholic) liver cirrhosis, liver steatosis, liver steatohepatitis, or some other pathologies? This point should be clarified, because it may affect data interpretation.
- Finally, the authors should thoroughly revise the manuscript and correct typing errors.
Discussion:
- “In case of lung cancer patient tissues, the higher p-PDHA1 revealed the much poorer survivability (data not shown).” The lung patient data should be shown in results, otherwise the reference to them should not be included in the Discussion.
- Discussion mentions the possibility that insulin may regulate the Warburg effect in HCC. Was lactate production increased by insulin treatments in in vitro experiments (in this study)?
- Given the amount of data presented in this manuscript, the discussion is somewhat underdeveloped. For instance, among other issues, the difference between mouse and human data could be examined in more detail.
- Experiments with the phosphomimetic mutant S293D appear to suggest that insulin treatment was needed for interaction with PKM2 (or at least increased the interaction). Since insulin is supposed to act via an increase in Ser293 phosphorylation, it would be important to discuss this point further.
- Discussion could also provide more references to the concept that metabolic enzymes perform multiple functions (in addition to their enzymatic function) since this study provides interesting new insights in this field.
Author Response
Comments and Suggestions for Authors
Hossain et al. provide an interesting study concerning novel roles of pyruvate dehydrogenase (PDHA1) and pyruvate kinase (PKM2). A major finding is that PDHA1, PKM2, and HIF-1α interact and are involved in regulation of LINC00273. This study certainly provides an interesting new insight into interaction between metabolic enzymes, HIF-1α, and histone acetylation. The authors performed an extensive amount of work; indeed, one of the major strengths of the study is the amount and versatility of the results, which range from cell cultures, including gene silencing and overexpression, to animal studies as well as assessment of clinicals samples and data.
Comments/questions:
- “Growth hormone (GH) such as insulin…” GH and insulin are two different hormones. The sentence should be revised.
- Thank you for your kind comment. We changed it as “Growth factor and insulin”.
- “Of interest, diabetes mellitus (DM) patients induced by insulin resistance and hyperinsulinemia…” This sentence should be revised so that it will describe the pathogenesis of (type 2) diabetes more precisely.
- We changed as follows: “Of interest, hyperinsulinemia and insulin resistance are associated with an increased risk of hepatocellular carcinoma (HCC).”
- Regulation of pyruvate dehydrogenase by metabolites should be at least mentioned so that it does not come across that phosphorylation is the only mechanism.Methods:
- General comment: to fully appreciate the data provided methods should be described in more detail. Please see specific comment below.
- Your comment is very constructive. Thus, we described as follows: “Furthermore, post-translational modification such as succinylation and metabolites including acetyl-CoA/CoA, NADH/NAD+ and ATP regulate PDC activity {Park, 2018 #1531}.”
- Cell culturing should be described more precisely: For instance, what was the glucose concentration in media? Were antibiotics/antimycotics and serum present in the media during experiments
- Thank you for your critical and careful comments: according to your suggestion we described briefly about cells culture in methods section. Almost all cancer cells line were maintained in Dulbecco’s modified Eagle’s medium (DMEM) containing high glucose 4500 mg/L D-glucose, L-glutamine, 110 mg/L sodium pyruvate and sodium bicarbonate and PC12 cells were maintained in RPMI 1640 medium 1X with low glucose 2000 mg/L, sodium bicarbonate 2000 mg/L, sodium chloride 6000 mg/L supplemented with 10% heat-inactivated fetal bovine serum (FBS) with 1% penicillin/streptomycin antibiotics. Cell culture flasks were maintained under 5% CO2, 95% ambient air and humidified conditions at 37°C. For experimental plate cells were first serum starvation for 12 h without serum and antibiotics and finally treated with insulin (100 nM) and others stimulant like LPS (10 ng/ml) and NGF (100 ng/ml). There is no serum and antibiotics in all experiment.
- Gene silencing should be described in more detail. For instance, 1) what was the concentration of siRNA, 2) at what point after addition of siRNA were experiments performed, 3) was there only one siRNA or was there a pool of siRNAs, etc.
- Thank you for your constructive comment. We described detail in 2.8 methods part. Transfection of Plasmid DNA and si-RNA: DNA and Small interfering RNA (si-RNAs) were transfected to cells using Lipofectamine 3000 (Invitrogen) and jetprime reagent according to the manufacturer's instructions. Various HA-pCMV-PKM2 domain and PDHA1 constructs were transfected by using Lipofectamine 3000 (Invitrogen) according to the manufacturer's instructions. In all transfection experiments, cells were first seeded and incubated for 8 h without antibiotics but containing growth media; then media were aspirated and serum free media without antibiotics was added for 12 h and insulin (100 nM) was added fresh serum free media into experimental plate, followed by 48 h of incubation for transfection. The media was then replaced by fresh full growth media with incubation for another 48 h before performing the stated experiments. In six well cells culture dish 2 µg DNA were transfected. Small interfering RNAs (si-RNA) si-PDHA1 (sc-91064 and ID: 18597), si-PKM2 (sc-62820) and control si-RNA (sc-37007) were purchased from Santa Cruz and Bioneer. These si-RNAs were transfected at a final concentration of 30 nM-50 nm for 48 h. si-LINC00273 was purchased from Bioneer and were transfected at a final concentration of 20-50 nM for 48 h. For double transfection first si-RNA was transfected then after 2 hr DNA was transfected into the cells according to the manufacture’s protocols. Sometimes we performed co-transfection for some experiments.
- Conditions under which PCR was performed should be described in more detail.
- Thank you for your careful comment. According to your opinion we briefly mention in 2.14 section. RNA isolation and Quantitative reverse transcriptase (RT-qPCR) to measure target genes mRNA level: Total RNA from HepG2 cells were isolated by using TRIzol reagent (Applied Biosystems/Ambion). Concentration and purity of the isolated RNA was measured by spectrophotometrically at 260 nm and 280 nm and by Nano Drop (Thermo Scientific). cDNA from RNA was made by reverse transcription with M-MLV reverse transcriptase (NEB-UK, Hitchin, UK). A mix of 2 μg total RNA, 1 μl Oligo-dT and 2 μl of LINC00273 primers (Bioneer) was incubated in a total volume of 10 μl for 5 min at 75°C and cooled to 4°C for 5 min in the PCR first reaction. To the mix, 2.5 μl of M-MLV 5× reaction buffer (NEB-UK), 5 μl dNTPs, 1 μl RNAase inhibitor (NEB-UK) and 1 μL of M-MLV reverse transcriptase were added to reach a total volume of 25 μl. The reaction mix was then incubated at 25°C for 10 min, 42°C for 60 min, 95°C for 10 min and then cooled to 4°C. For the cDNA reaction mix, PCR reactions were then performed using rTaq 5x Master Mix (ELPIS-Biotech, Daejeon, Korea) in reactions containing 10 pM of each primer, 4 μl cDNA, and 10x PCR buffer as supplied by the manufacturer, in a total volume of 20 μl. PCR primers of GAPDH) were used for analysis as house-keeping control gene. The GAPDH PCR reaction were run at 95°C for 5 min, 20 cycles of 95°C for 15 sec, 60°C for 20 sec, 72°C for 20 sec, concluded with 72°C for 5 min and cooled to 4°C.The cDNA was then used with the ExcelTaq SYBR Rox 2X fast Q-PCR master mix (TQ1210, SMOBIO Technology. Inc, Taiwan) for real-time qPCR (RT-qPCR) quantification using an Applied Biosystems Step one plus PCR system. All samples were analyzed in triplicate. Relative quantities of specifically amplified cDNA were determined using the comparative threshold cycle (CT) values method. GAPDH was used as an endogenous reference gene, and without template and reverse-transcription controls were used to exclude nonspecific amplification. For measuring LINC00273 mRNA levels, primer sequences were designed and PCR reaction conditions were: 95°C for 5 min, followed by 30 cycles of 95°C for 30 sec, 54°C for 30 sec, 72°C for 30 sec, concluded by 72°C for 5 min and cooled to 4°C. Products were run on 1% agarose gels and visualized by ethidium bromide staining. RT-PCR products were run on 1% agarose gels and visualized by ethidium bromide staining. Primers were designed for GAPDH: forward, 5’-AGAAGGCTGGGGCTCATTTG-3’, reverse, 5’-AGGGGCCATCCACAGTCTTC-3’; for LINC00273: forward, 5’-GCCACACAGTAGGTGACGAG-3’, reverse, 5’-ACTGCTTTCGGGAGAGAATG-3’; for GPR174: forward, 5’-TGTGCCAGGTCTCATAGGGA-3’, reverse, 5’-AGTCATGGAAGCGAAAGGGG-3; for KDM1B: forward, 5’-GAGGGACAGGTGCTTCAGTT-3’, reverse, 5’-CACTGCACTGGAGATTTGAG-3’.
- Several animal models were used in the study: which ethics approval refers to the insulin treatment in rats (paragraph 2.16)?
- Thank you for your careful observation. This study was conducted in accordance with the strict guidelines of the institutional Animal Studies Care and Use committee of the Hallym University in Chuncheon, Korea (protocol no. Hallym 2017-3). Animal sacrificed was carried out using isoflurane anesthesia, and we attempted to reduce minimal pain and distress.
- The method by which experimental animals were killed should be reported. Also, what kind of chow did they have, were they fasted or not? If yes, for how long, etc.
- Thank you for your critical observation. We follow this method for animals kill. Mice were killed by urethane solution injection into the abdominal cavity. For in this study we collected BALB/c (female, 4–6 weeks old) and C57/BL-6J (female 4-6 weeks old, 15-17 g) mice were obtained from Samtako (Osan, Korea). These all mice were normal mice not nude mice. we did not use any chow model mice for tumor growth and these mice were not fasted in total xenograft experiment for this study.
- Quantification of immunocytochemistry data should be described in more detail.
- Thank you for your valuable comments. According to your suggestion briefly we described in 2.11 section. HepG2 Cells were cultured and fixed in 4% paraformaldehyde for 10 min and for membrane permeabilization we used 1XPBST (containing with 0.1% TX-100 detergents and PBS) for 10 min then washed with 1XPBS and incubated with specified primary antibody (1:100) overnight at 4°C. After primary antibody incubation again washed with 1XPBS then p-PDHA1 and PKM2 antibody recognized by an Alexa Fluor 488-conjugated secondary antibody (green-color emission) and Alexa-546 conjugated anti-rabbit IgG (red-colour emission, Molecular Probes) with 1:50 dilution for 2 h at room temperature. DAPI (1 μg/ml) was also added 10 min before washing to label the nuclei [16]. For dual immunofluorescence, either monoclonal anti-p-PDHA1 or anti-PKM2 at 1:100 dilution was added and incubated with the cells overnight at 4oC. Then added Alexa-488 conjugated anti-mouse IgG (green-colour emission, Molecular Probes) with 1:50 dilution and Alexa-546 conjugated anti-rabbit IgG (red-colour emission, Molecular Probes) with 1:50 dilution respectively. The nuclear region was stained with DAPI (4',6-diamidino-2-phenylindole). Fluorescence images were obtained with a conventional fluorescence microscope (Axiovert 200, Zeiss, Oberkochen, Germany). We measured the cyan color intensity in nucleus by using adobe Photoshop version 7 and plotted the relative intensity as a bar diagram.
- Description of clinical characteristics of patients included in the study is insufficient.
- Thank you for your comment. We already prepared all patients information according to Korean Statistical Information Service KOSTAT database (KOSIS). We presented this data into (sFig.2A-2B).
- A. Data of liver cancer patients B. Data of lung cancer patients
- Xenograft experiment should be described in more detail.
- Thank you for your careful observation. As you mention that we must provide details information about xenograft experiment. Here, briefly we described in 2.17 methods section. For the murine tumor model, BALB/c (female, 4–6 weeks old) and C57/BL-6J (female 4-6 weeks old, 15-17 g) mice were obtained from Samtako (Osan, Korea). The animal experiment protocol in this study was reviewed and approved by the Hallym University-Institutional Animal Care and Use Committee (IACUC) (Hallym 2021-6). Other protocol was similar to the previous report [16]. We did not use any chow model mice for tumor growth, and they were not fasted. For tumor implantation, non-transfected and transfected 4T1 cells were washed twice by serum-free culture solution and 1x107 cells (100 µl in PBS) were subcutaneously injected into the right flank of each mouse. When tumor volumes reached 100 mm3, tumors size measures were made three times a week using Vernier calipers. Volume (mm3) of tumors was calculated using the standard formula of length × width2 and growth curves were drawn by the Prism software (GraphPad). For tumor induction, we applied daily phorbol 12-myristate 13-acetate (PMA) solution (10 μl of 1 μM) near the tumor implanted area. The engrafted tumors were monitored for 28 days in case of 4T1 cells but for Hepa1c1c7 cells only 20 days and mice were then sacrificed, and their tumor samples were harvested. Mice were killed by urethane solution injection into the abdominal cavity. We attempted to reduce minimal pain and distress.
- Methodology of measuring tumours in mice in vivo should also be described.
- Thank you for valuable comments. When tumor volumes reached 100 mm3, tumors size measures were made three times a week using Vernier calipers. Volume (mm3) of tumors was calculated using the standard formula of length × width2 and growth curves were drawn by the Prism software (GraphPad). For tumor induction, we applied daily phorbol 12-myristate 13-acetate (PMA) solution (10 μl of 1 μM) near the tumor implanted area. The engrafted tumors were monitored for 28 days in case of 4T1 cells but for Hepa1c1c7 cells only 20 days and mice were then sacrificed, and their tumor samples were harvested. Mice were killed by urethane solution injection into the abdominal cavity.
Results:
- General comments: It would be better if immunoblots were quantified and statistically analysed wherever possible. This should be done at least for the most important results.
- You are right. As you suggested, we supplemented statistic results in many figures (Fig. 1F, Fig. 1H, Fig. 1K, Fig. 1N, Fig. 2A, Fig. 2C, Fig. 2D, Fig. 2E, Fig. 3C, Fig. 3D, Fig. 3F, Fig. 3G, Fig. 3H, Fig. 3I, Fig. 4A, Fig. 4B, Fig. 4C, Fig. 5A, Fig. 6A, Fig. 6M, Fig. 6N, Fig. 6O and Fig. 6P).
- The presence of serum may affect the results of insulin treatment. At least for a few critical experiments, the authors should show that the classical insulin pathway was activated (e.g. Akt) by insulin.
- This is very constructive comment. As you suggested, we confirmed that insulin induced p-AKT in HepG2 cells (Fig. 1E). We also observed p-AKT in HepG2 cells stimulated by insulin several times before {Islam, 2019 #810}.
- Fig. 1B: what was the insulin concentration and how long was the treatment? Insulin appears to have stimulated interaction with the phosphomimetic mutant, but not control cells. Could authors comment this result?
- Insulin was treated at 100 nM for 48h. Thank you for your kind comment. We described the sentences as follows:
“Indeed, GST-PDHA1 S293D, a phosphomimetic form strongly interacted with PKM2 in the lysates of HepG2 cells, which were stimulated with insulin, but not with the PKM2 in control cell lysats (Fig. 1B). This suggests that additional factor(s) stimulated by insulin may be involved in p-PDHA1 and PKM2 interaction, for instance additional post-translational modification(s) of either PDHA1 or PKM2.”
- Fig. 1I: What was the insulin concentration and how long was the treatment?
- Insulin was treated at 100 nM for 48 h.
- Fig. 1M: For how long were cells treated with LPS?
- LPS was treated at 10 μg/ml for 4 h.
- As mentioned above, quantification (densitometry) should be performed for data in Fig. 1. Also, the total PDHA1 should be included as a control in all blots where p-PDHA1 was assessed.
- As you suggested, we supplemted total PDHA1 levels and quantified the data.
- Fig. 2: Scale bars are missing from lower magnification micrographs.
- Scale bar: 40 μm
- Fig. 2C requires a control blot – total PDHA1.
- As you suggested, we added total PDHA1 blot as control.
- Fig. 2: How effective was gene silencing of PKM2 and PDHA1? A control experiment showing the effectiveness of gene silencing of PKM2 and PDHA1 should be shown (at least immunoblot).
- Thank you for your very kind comments. As you suggested, we added immunoblot result in Fig. 2D and 2E
- Fig. 2: MTT assay is an indirect measure of cell proliferation: i.e. it does not assess cell proliferation directly. How was cell proliferation assay performed (how many cells were seeded, what was media composition – serum, antibiotics, antimycotics)?
- We described the Methods: “HepG2 cells were seeded either in 12-well (1x105 cells/well) or 6-well dishes (4x105 cells/well) and 96 well plate (1x103 cells/ well). Cells were serum-starved for 12 h before adding 100 nM insulin. These media contain high glucose 4.5 g/L, L-gultamine, 110 mg/L sodium pyruvate and sodium carbonate without serum and antibiotics.”
- Fig. 3: It would be important to assess proliferation (MTT assay) of cell cultured treated with si-LINC00273 in the absence of insulin. This is an important control experiment.
- As you commented, we added si-LINC00273 control without insulin (Fig. 3D).
- Fig. 3: How effective was silencing of LINC00273? It would be important to show that its knock-down was effective by performing PCR and showing quantified results. Similarly, silencing of PDHA1 and PKM2 should be assessed quantitatively. While Fig. 4 shows effective silencing in insulin-treated cells, at least a few controls should be shown for results in Fig. 2 and 3 (blots or immunocytochemistry, depending on the type of analysis). •Thank you for critical comments. As you suggested we quantified LINC00273 by si-LINC00273. In addition, we added quantified LINC00273, p-PDHA1 and PKM2 levels.
- Fig. 2 and 3: Agarose gel electrophoresis was used to quantify RNA. However, a real-time PCR would provide a more precise (more quantitative) result.
- As you commented, we measured LINC00273 levels by real-time qPCR (Fig. 3H and 3I).
- Fig. 4 and 6: Clinical correlations are interesting and relevant; however, their usefulness is limited if patients’ characteristics are not indicated. Were patients in Fig. 4G and H in the same cancer stage? Could the stage have affected ZEB expression and therefore confounded the results in Fig. 4H?
- Unfortunately, we could not distinguish the cancer stages in the reference data. This data obtained from (http://ulcan.path.uab.edu/) database. Survival probabilities for low and high ZEB1 expression in several cancer types were obtained and plotted via Kaplan-Meier plotter [Pan-cancer RNAseq, TCGA survival]
- Also, is there any difference between cancer patients in Fig. 6F, H, and J who survived and those who did not survive? Did those who died have a higher cancer stage etc.? Did phosphorylation differ between stages (e.g. was phosphorylation higher in those with a higher stage?) or did phosphorylation correlate with a poorer prognosis for patients with the same cancer stage?
- Unfortunately, we could not distinguish the cancer stages in the (reference data) Information on death and survival of the patients were obtained from the Korean Statistical Information Service KOSTAT database (KOSIS) of the Korean government. which lacked the detail information about liver cancer stages of patients (Fig. 6E-6K and sFig. 3A). However, fortunately we could get cancer stage information of lung cancer patients (Fig. 6L-6R and sFig. 3B).
- To what time after diagnosis does the “live/dead” status of patients refer to? Did patients die of cancer or did perhaps some die of other causes?
- As you commented, we added the patient information of the date when they were diagnosed and die (Supplementary Fig. 3A-3B).
- Also, I suggest the authors to consider different terminology with regard to the patients – “live/dead status” is usually used for cell culture and may not be the most appropriate choice of words for patients.
- Thank you for your kind comment. We used “Survival patient’s and Non-survival patient’s”.
- Fig. 5: The text and/or figure legend should always make absolutely clear which cells were used for experiments and what were the treatments (insulin concentration, duration of treatment etc.).
- As you commented, we revised figure legend in Fig.5.
- Fig. 5A: There appears to be less p-PDHA1 upon insulin treatment (IP results)? What does the quantification and statistical analysis indicate?
- We prepared statistic result and found p-PDHA1 and PKM2 binding to STAT3 increased by insulin (Fig. 5A).
- Fig. 6: siPDHA1 and siPKM2 cells were used for the xenograft experiment; however, how does silencing of PDHA1 and PKM2 affect the in vitro viability of Hepa1c1c7 and 4T1 cells? Were they capable of proliferating normally, was their survival reduced/apoptosis increased? These aspects require a more detailed assessment because they are important for interpretation of the in vivo results.
- Based on your critical comment, we conducted new experiments. Si-PDHA1 and reconstituted S293A PDHA1 reduced cell proliferation in 4T1 cells upon insulin (Fig. 6A).
- Fig. 6: Were insulin concentrations measured in patients with HCC? If not, the reference to possible effect of different insulin concentrations should be omitted from the Results.
- We did not measure insulin concentrations of HCC patients. As you commented, we omitted reference from the results. Of interest, we found a paper that insulin plasma level in HCC (18.2.+/- 18.8) is higher than chronic hepatitis C (11.5 +/- 10.5) and liver cirrhosis (17.6+/- 11.2) patients (Reference: Clinical Medicine: Endocrinology and Diabetes 2009:2, 25-33; Table 1, p=0.001).
- HCC has different underlying causes: what kind of patients were included in the study: those with type 2 diabetes, HCV infection, alcoholic (or non-alcoholic) liver cirrhosis, liver steatosis, liver steatohepatitis, or some other pathologies? This point should be clarified, because it may affect data interpretation.
- We agree with you. However, we are sorry for the deficiency of those information of liver cancer patients. When we requested and obtained the liver cancer tissues form cancer center at that time, we realized the only the information of prepared chart in sFig. 3A.
Discussion:
- “In case of lung cancer patient tissues, the higher p-PDHA1 revealed the much poorer survivability (data not shown).” The lung patient data should be shown in results, otherwise the reference to them should not be included in the Discussion.
- We presented lung cancer patients data in Fig. 6L-6P and described their contents in the results section.
- Discussion mentions the possibility that insulin may regulate the Warburg effect in HCC. Was lactate production increased by insulin treatments in in vitro experiments (in this study)?• It is a very nice question to consider. We speculated the Warburg effect by an increase of p-PDHA1 in response to insulin. We did not conduct the experiment to measure lactate level upon insulin. However, insulin induced lactate production in HepG2 cells and Bel7402 cells (Fig. 1e) as well as glucose consumption (Fig. 1d), which was published in “Oxidative Medicine and Cellular Longevity 2014: Article ID 504953 (2014), http://dx.doi.org/10.1155/2014/504953.” Thereby, we described the sentence as follows in Discussion section: “Indeed, insulin was reported to increase lactate level in HepG2 and Bel7402 HCC cells in vitro {Li, 2014 #1532}”
- Given the amount of data presented in this manuscript, the discussion is somewhat underdeveloped. For instance, among other issues, the difference between mouse and human data could be examined in more detail.
- In addition, we described also the sentences as follows in Discussion section: “However, the high and low levels of p-PDHA1 did not distinguished difference in survival probability contrary to our expectation (Fig. 6G). This may be due to the different transient insulin concentrations in collected blood samples form liver cancer patients, as insulin sensitively augmented both PKM2 and p-PDHA1 levels in hepatocellular carcinoma cell lines (Fig. 1E-1G).”
- • As you suggested, we discuss more about mouse xenograft experiment (Fig. 6B-6D) and human survival graph (Fig. 6F and 6G). It was difficult for us to interpret the result. In case of lung cancer, the patients containing the higher p-PDHA1 revealed the poorer survival probability (New results: Fig. 6L-6P). We previously described the follows in result section. “Contrary to our expectation, p-PDHA1 did not display differential survival for the liver cancer patients analysed likely due to alteration of endogenous insulin concentration in patients (Fig. 6F and 6G).”
- Experiments with the phosphomimetic mutant S293D appear to suggest that insulin treatment was needed for interaction with PKM2 (or at least increased the interaction). Since insulin is supposed to act via an increase in Ser293 phosphorylation, it would be important to discuss this point further.
- Thank you for a constructive comment. We described the sentences in Discussion section as follows: “The structure of PDHA1 phosphorylated at Ser293 residue may be different from the dephosphomimic-form. Herein p-Ser293 of PDHA1 may facilitate its binding to CTD of PKM2 (Fig. 1J).”
- Discussion could also provide more references to the concept that metabolic enzymes perform multiple functions (in addition to their enzymatic function) since this study provides interesting new insights in this field. •We carefully revise and we asked professional revision from editing company.
- Finally, the authors should thoroughly revise the manuscript and correct typing errors.
- Thank you for a constructive comment. Based on your comment, we described the sentences as follows. “Notably, a large number of metabolic enzymes have noncanonical or nonmetabolic functions. Of note, several metabolic enzymes such as ketohexokinase isoform A (KHK-A), hexokinase (HK), nucleoside diphosphate kinase 1 and 2 (NDP kinase1/2: NDKP1/2), and phosphoglycerate kinase 1 (PGK1) as well as PKM2 are also able to function as protein kinases that phosphorylate a variety of protein substrates to regulate the Warburg effect, gene expression, cell cycle progression and proliferation and many other underlying cellular functions {Lu, 2018, 29463470; Xu, 2021 #1533}. For instance, PGK1 localized in cytosol, nucleus or mitochondria regulates specific cellular functions. Mitochondrial PGK1 plays a role as a protein kinase, which phosphorylates pyruvate dehydrogenase kinase 1 (PDHK1) at Thr338, thereby leading to PDHK1 activation. Then the active PDHK1 phosphorylates PDHA1 at Ser293, ensuring PDH inactivation, lactate production and brain tumorigenesis {Li, 2016, 26942675}.”

Reviewer 2 Report
Immortal hepatocyte cell lines are widely used to elucidate insulin-dependent signaling pathways and regulation of hepatic metabolism. However, the often tumorigenic origin might not represent the metabolic state of healthy hepatocytes. Concerning gluconeogenesis and hepatokine expression, HepG2 cells appear to be closer to the in vivo situation despite the tumorigenic origin.
The authors investigated in the current study a mechanism by which insulin regulates gene expression through p-PDHA1 and PKM2 in hepatocarcinoma cells. The authors noticed that "diabetes mellitus (DM) induced by insulin resistance and hyperinsulinemia are associated with an approximately 2-3-fold increased risk of hepatocellular carcinoma".
But, there are serious concerns about this hypothesis. First of all, when DM develops, 50% of insulin secretory capacity from β- cells is lost. So, patients with DM have hypo-insulinemia in comparison with healthy people. The milieu outside the hepatic cell in DM is characterized by hyperglycemia, elevated free fatty acids, and decreased insulin levels. The liver's inappropriate insulinism has been considered a primary pathogenetic mechanism for DMT2. So, I think that the researchers should do the experiment in a milieu of elevated glucose and FFA to investigate the action of insulin.
Therefore there is a problem with the methodology.
But, the manuscript could be published if the authors clarify the milieu of the experiment and not only insulin. It will be helpful for other future investigators. And the authors should make corrections in the introduction section and add a paragraph with the limitations noticed above.
Author Response
Immortal hepatocyte cell lines are widely used to elucidate insulin-dependent signaling pathways and regulation of hepatic metabolism. However, the often tumorigenic origin might not represent the metabolic state of healthy hepatocytes. Concerning gluconeogenesis and hepatokine expression, HepG2 cells appear to be closer to the in vivo situation despite the tumorigenic origin.
The authors investigated in the current study a mechanism by which insulin regulates gene expression through p-PDHA1 and PKM2 in hepatocarcinoma cells. The authors noticed that "diabetes mellitus (DM) induced by insulin resistance and hyperinsulinemia are associated with an approximately 2-3-fold increased risk of hepatocellular carcinoma".
But, there are serious concerns about this hypothesis. First of all, when DM develops, 50% of insulin secretory capacity from β- cells is lost. So, patients with DM have hypo-insulinemia in comparison with healthy people. The milieu outside the hepatic cell in DM is characterized by hyperglycemia, elevated free fatty acids, and decreased insulin levels. The liver's inappropriate insulinism has been considered a primary pathogenetic mechanism for DMT2. So, I think that the researchers should do the experiment in a milieu of elevated glucose and FFA to investigate the action of insulin.
It is constructive comments. However, at this moment, we observed, the novel difference of normal liver cells and HepG2 hepatocellular carcinoma cells in insulin signaling pathway, as we know, insulin reduced p-Ser293 PDHA1 levels in normal rat liver tissue, but it increased p-Ser293 levels in HepG2 cells. Therefore, we focused on the p-PDHA1 increase induced by insulin in HepG2 cells, not DM itself.
In our current study, we have the limitation that we didn’t use any diabetic/hepatocellular carcinoma clinical patient samples for further study the role p-PDHA1/PKM2 complex in HCC progression. Instead, we utilized hepatocellular carcinoma model cell lines including HepG2 and Hepa1c1c7 cells. In addition, we focused insulin action regarding PDHA1 Ser293 phosphorylation, because p-PDHA1 was increased in HepG2 cells in contrast the normal liver tissue, in which p-PDHA1 was decreased by insulin. In addition, we found p-PDHA1 interacts with /PKM2 forming a complex, which exits in nucleus and regulates histone acetylation and specific gene expression. We believed that these phenomena are involved in HCC progression and it will be conducive for future investigators. However, the clinical and pathological studies of p-PDHA1/PKM2 complex in different cancer patients are poorly understood and the further clinical study about this will be required in the future.
We consider that the introduction in our manuscript may mislead other researchers about the concept of our research consisting of biochemical and cellular biology results including p-PDHA1/PKM2 interaction, nuclear translocation and the regulation of histone acetylation. Therefore, we changed the description in Introduction as follows:
“Recently, we reported that insulin induces an increase in p-Ser293 PDHA1 along with p-Ser473 Akt levels in HepG2 hepatocellular carcinoma cells. However, insulin reduced p-Ser293 PDHA1 in insulin-treated rat liver tissue and rat primary hepatocyte {Islam, 2019, 30226812}. However, the role of p-PDHA1 in hepatocellular carcinoma HepG2 cells due to insulin is poorly understood. Herein, it was necessary to study the effect of insulin on HCC cells such as HepG2 cells as a model system with regard to PDHA1 Ser293 phosphorylation.”
Therefore there is a problem with the methodology.
Thank you for your valuable observation: we corrected our methodology and described in detail in materials and method section.
But, the manuscript could be published if the authors clarify the milieu of the experiment and not only insulin. It will be helpful for other future investigators. And the authors should make corrections in the introduction section and add a paragraph with the limitations noticed above.
- Thank you for your kind and critical comments. We thought type 2 DM containing hyperinsulinemia and insulin resistance may be associated with a high risk of HCC in the beginning. Thus, we described the relationship between DM and cancer in Discussion part as follows:
“Of interest, hyperinsulinemia and insulin resistance may be associated with an increased risk of hepatocellular carcinoma (HCC) {Li, 2017 #1448}. DM2 was reported to be associated with a three-fold increased risk of heparocarcinoma occurrence {Donadon, 2008 #1534}. Hereby, it was necessary to study the effect of insulin on HCC cells. However, we have the limitation that we didn’t use any diabetic/hepatocellular carcinoma clinical patient samples in this study. Therefore, it remains to be elucidated in clinical further study that p-PDHA and PKM2 complex in the HCC patients is crucial for the HCC progress.”
- Of interest, we found a paper that insulin plasma level in HCC (18.2.± 18.8) is higher than chronic hepatitis C (11.5 ± 10.5) and liver cirrhosis (17.6 ±11.2) patients (Reference: Clinical Medicine: Endocrinology and Diabetes 2009:2, 25-33; Table 1, p=0.001).

Reviewer 3 Report
The manuscript, which title is Pyruvate dehydrogenase A1 phosphorylated by insulin associates with pyruvate kinase M2 and induces LINC00273 through histone acetylation, is novelty and interesting. However, the authors should provide the limitation of study in the manuscript. And then, the materials should show the same format. Authors should provide high resolution of figure.
Author Response
The manuscript, which title is Pyruvate dehydrogenase A1 phosphorylated by insulin associates with pyruvate kinase M2 and induces LINC00273 through histone acetylation, is novelty and interesting. However, the authors should provide the limitation of study in the manuscript. And then, the materials should show the same format. Authors should provide high resolution of figure.
1, Thank you for your comments. As you suggested, we described the sentences as follows in Discussion section: “However, it remains to be elucidated in clinical further study that p-PDHA1 and PKM2 complex in the HCC patients was crucial for the HCC progress.”
- We made the same format of the materials.
- We provide the high resolution figures in this time.

Round 2
Reviewer 2 Report
no-other comments